# Affine Invariance in Continuous-Domain Convolutional Neural Networks

## Abstract

The notion of group invariance helps neural networks in recognizing patterns and features under geometric transformations. Group convolutional neural networks enhance traditional convolutional neural networks by incorporating group-based geometric structures into their design. This research studies affine invariance on continuous-domain convolutional neural networks. Despite other research considering isometric invariance or similarity invariance, we focus on the full structure of affine transforms generated by the group of all invertible $2 \times 2$ real matrices (generalized linear group $\mathrm{GL}_2(\mathbb{R})$). We introduce a new criterion to assess the invariance of two signals under affine transformations. The input image is embedded into the affine Lie group $G_2 = \mathbb{R}^2 \rtimes \mathrm{GL}_2(\mathbb{R})$ to facilitate group convolution operations that respect affine invariance. Then, we analyze the convolution of embedded signals over $G_2$. In sum, our research could eventually extend the scope of geometrical transformations that usual deep-learning pipelines can handle.

## 1 Introduction

Convolutional neural networks (CNNs) have achieved strong performance across a wide range of applications by exploiting translation equivariance and local structure (LeCun et al., 1998; Krizhevsky et al., 2012). However, their ability to handle more general geometric transformations remains limited (Cohen & Welling, 2016). In many practical settings, such as imaging, remote sensing, and pattern recognition, data are subject to affine transformations, combinations of linear mappings and translations, which introduce significant geometric variability while preserving structural properties such as parallelism.

Group convolutional neural networks (G-CNNs) extend classical CNNs by incorporating symmetry structures through group actions, enabling equivariance to transformations beyond translation (Cohen & Welling, 2016). By operating over transformation groups, G-CNNs can capture geometric features such as orientation, scale, and position, reducing the need for extensive data augmentation and improving robustness. Over time, G-CNNs have been developed for discrete (Cohen & Welling, 2016; Winkels & Cohen, 2018; Dieleman et al., 2016; Worrall & Brostow, 2018; Hoogeboom et al., 2018), continuous (Oyallon & Mallat, 2015; Bekkers et al., 2015; Weiler et al., 2018; Zhou et al., 2017), and steerable variants (Cohen et al., 2018; Worrall et al., 2017; Kondor & Trivedi, 2018; Thomas et al., 2018; Andrearczyk et al., 2019). Extensions include separable convolutions for scale-rotation-translation equivariance (Knigge et al., 2022) and efficient implementations of SE(3) equivariance (Chen et al., 2021).

Despite these advances, modeling general affine transformations remains challenging. Many existing approaches focus on restricted transformation groups or rely on continuous formulations that are difficult to implement in practice. For instance, (MacDonald et al., 2022) propose Monte Carlo integration over arbitrary Lie groups, while (Li et al., 2024) develop affine-equivariant networks based on differential invariants. Although these methods provide strong theoretical foundations, they often involve complex constructions or specialized feature representations.

In this work, we study affine invariance in continuous-domain convolutional neural networks, focusing on transformations generated by the general linear group $\mathrm{GL}_2(\mathbb{R})$. Our approach follows a three-layer lifting–

convolution–projection architecture. Specifically, the input signal is first embedded into the affine Lie group $G_2 = \mathbb{R}^2 \rtimes \mathrm{GL}_2(\mathbb{R})$, followed by convolution in this space, and finally projected back to the original domain.

Intuitively, the proposed approach approximates affine equivariance by explicitly transforming the input signal under a finite set of affine transformations and processing these transformed versions using shared convolutional operators. By aggregating the resulting feature maps through a projection step, the network captures transformation-invariant information while retaining discriminative spatial structure. This design provides a practical alternative to continuous group formulations, enabling affine-aware representations without requiring complex group integrations or specialized invariant constructions.

We show that this architecture yields stable representations under affine transformations and that the associated computations can be simplified from group convolutions to integrals over Euclidean space. In addition to the theoretical analysis, we provide an explicit and implementable construction of the proposed architecture and demonstrate its effectiveness in practice.

Motivated by the limitations of standard convolutional architectures in handling general affine transformations, we aim to develop a framework that explicitly incorporates affine structure into the network design. In particular, rather than relying on data augmentation to learn invariance, we construct representations that are inherently stable under affine transformations. This approach enables improved generalization in settings with significant geometric variability, especially when training data is limited.

Our main contributions are as follows:

- We analyze affine invariance in continuous-domain G-CNN architectures generated by $GL_2(\mathbb{R})$.

- We develop a lifting–convolution–projection framework and show its stability under affine transformations.

- We provide a formulation that reduces group convolution over the affine group to tractable integrals over Euclidean space.

- We demonstrate empirically that the proposed model improves robustness to affine distortions compared to a standard CNN baseline.

The following section introduces the necessary group-theoretic background, notation, and illustrative examples that form the foundation for the subsequent analysis.

## 2    Preliminaries

Ensuring the equivariance of artificial neural networks (NNs) with respect to a group $G$ is an essential characteristic, as it guarantees that applying transformations to the input preserves all information, merely shifting it to different network locations. It has been determined that when aiming for equivariant NNs, the sole viable choice is to employ layers in which the linear operator is defined through group convolutions. We begin by examining what a neural network framework for continuous signals looks like. For vector spaces, the most general form of a linear transformation is a matrix–vector multiplication, while for continuous signals, linear transformations are expressed through kernel operators. A neural network framework for handling continuous signals can be formulated using the following equation:

$$\boldsymbol{y} = \sigma\left(\mathcal{K}\Phi(\boldsymbol{x})\right), \tag{1}$$

where $\boldsymbol{x} \in \mathcal{X}$ represents the input vector, and $\Phi(\boldsymbol{x})$ denotes the input signal. For example in an image input, $\boldsymbol{x}$ is the location of pixels and $\Phi(\boldsymbol{x})$ is the value that image takes in each pixel. Moreover, $\mathcal{K} : \mathbb{L}_2(\mathcal{X}) \to \mathbb{L}_2(\mathcal{Y})$ denotes a linear map, and $\sigma$ is the activation function. The kernel operator $\mathcal{K}$ is also defined as follows

$$\Phi^{l+1} = \mathcal{K}\Phi^l = \int_{\mathcal{X}} K(\boldsymbol{x}, \boldsymbol{y})\Phi^l(\boldsymbol{x})\mathrm{d}\lambda_{\mathcal{X}}(\boldsymbol{x}),$$

where $\mathcal{K} : \mathbb{L}_2(\mathcal{X}) \to \mathbb{L}_2(\mathcal{Y})$, $d\lambda_{\mathcal{X}}$ is a Radon measure on $\mathcal{X}$, $K(\boldsymbol{x}, \boldsymbol{y})$ denotes the kernel function, and $\Phi^l \in \mathbb{L}_2(\mathcal{X})$ denotes the feature map at layer $l$, i.e., a square-integrable function that serves as the input to

the next layer in a continuous neural network. To broaden the application of this explanation to the notion of group convolutional neural networks, we revisit a number of crucial definitions.

**Definition 1** (Group). *A group $(G, \cdot)$ is a set $G$ equipped with a binary operator represented by a dot symbol. The dot operator is associative $(g_1 \cdot g_2) \cdot g_3 = g_1 \cdot (g_2 \cdot g_3)$, has an identity element $e$. Moreover, every element of the set has an inverse element $g \cdot g^{-1} = g^{-1} \cdot g = e$.*

In our considerations, the set comprises functions, such as translations or rotations. The group operation operates on elements of this set through addition or multiplication Herstein (1991). We also need to define normal groups. A normal group is a subgroup $N$ of a group $G$ such that, for every element $g$ in $G$, the conjugate $gNg^{-1}$ is contained within $N$.

**Example 1** (Translation group). *The translation group in $\mathbb{R}^2$ is denoted by $(\mathbb{R}^2, \cdot)$ consists of all possible translations and is equipped with the below group product and group inverse:*

$$g \cdot g' = (\boldsymbol{x} + \boldsymbol{x}')$$
$$g^{-1} = -\boldsymbol{x},$$

*where $g = (\boldsymbol{x})$ and $g^{-1} = (-\boldsymbol{x})$ and $\boldsymbol{x}, \boldsymbol{x}' \in \mathbb{R}^2$.*

One important example of groups are Lie groups, which are defined as follows:

**Definition 2** (Lie groups). *A Lie group is a set $G$ with two structures: $G$ is a group and $G$ is a (smooth, real) manifold. These structures agree in the following sense: multiplication and inversion are smooth maps.*

Roto-translation symmetries of Euclidean spaces are examples of Lie groups, which is explained in the next example.

**Example 2** (Roto-translation group). *The roto-translation group in $\mathbb{R}^2$ is denoted by $\mathrm{SE}(2)$. The group $\mathrm{SE}(2) = \mathbb{R}^2 \rtimes \mathrm{SO}(2)$ consists of translations vectors in $\mathbb{R}^2$, and rotations in $\mathrm{SO}(2)$ and is equipped with the group product and group inverse:*

$$g.g' = (\boldsymbol{x}, \boldsymbol{R}_\theta) \cdot (\boldsymbol{x}', \boldsymbol{R}_{\theta'}) = (\boldsymbol{R}_\theta \boldsymbol{x}' + \boldsymbol{x}, \boldsymbol{R}_{\theta+\theta'})$$
$$g^{-1} = (-\boldsymbol{R}_\theta^{-1} \boldsymbol{x}, \boldsymbol{R}_\theta^{-1}),$$

*for $g = (\boldsymbol{x}, \boldsymbol{R}_\theta)$, $g' = (\boldsymbol{x}', \boldsymbol{R}_{\theta'})$, $\boldsymbol{R}_\theta = \begin{pmatrix} \cos\theta & -\sin\theta \\ \sin\theta & \cos\theta \end{pmatrix}$, and identity element $(\boldsymbol{0}, \boldsymbol{I})$. Note that $\rtimes$ denotes the semidirect product. In a direct product $G = H \times K$, both $H$ and $K$ are normal in $G$. Semidirect products are a relaxation of direct products where only one of the two subgroups must be normal.*

The group operator provides instructions on how to act on the group elements, ensuring that the result remains within the group. Of particular interest are symmetry groups, where each element in the set represents a symmetry transformation. When the group acts on a specific space, it is referred to as a group action.

**Definition 3** (Group action). *Let $\chi$ be a set. If $G$ is a group with identity element $e$, then a group action $\alpha$ of $G$ on $\chi$ is a function, $\alpha : G \times \chi \rightarrow \chi$, (which is usually denoted as $\alpha(h, \boldsymbol{x}) = h \odot \boldsymbol{x}$) that satisfies identity and compatibility conditions ($e \odot \boldsymbol{x} = \boldsymbol{x}$, $g \odot (h \odot \boldsymbol{x}) = (g \cdot h) \odot \boldsymbol{x}$) for all $g, h \in G$ and all $\boldsymbol{x} \in \chi$.*

For example the action of group $G = \mathrm{SO}(d)$ on space $\chi = \mathbb{R}^d$ could be denoted by $g \odot \boldsymbol{x} = \boldsymbol{R}\boldsymbol{x}$, where $\boldsymbol{x} \in \mathbb{R}^d$ and $\boldsymbol{R} \in \mathrm{SO}(d)$. For the set of points, we perform transformation through group products, while in the convolution kernel, we perform transformation via group representations. Therefore we need to understand representations. The multiplication within a group instructs us on merging transformations, yet it does not provide guidance on utilizing these transformations on other entities like vectors or signals. To address this, we require the concept of group action and group representations. Nevertheless, frequently, our attention is predominantly directed towards linear group actions operating on vector spaces, and these actions are termed representations.

**Definition 4** (Representation). *A representation is an invertible linear transformation $\rho(g) : V \to V$ parameterized by a group elements $g \in G$ that acts on some vector space $V$, which follows the group structure (it is a group homomorphism) via*

$$\rho(g)\rho(h)v = \rho(g \cdot h)v$$

*for $v \in V$ and $g, h \in G$.*

A standard approach for defining the action of a group $G$ on functions is via the regular representation.

**Definition 5** (Regular representation). *Let $\Phi \in \mathbb{L}_2(\mathcal{X})$. Then the regular representation of $G$ acting on $\mathbb{L}_2(\mathcal{X})$ is given by*

$$\rho(g)\Phi(\boldsymbol{x}) = \Phi\left(g^{-1}\boldsymbol{x}\right).$$

**Example 3** (Regular representation of roto-translation group). *Let $\Phi \in \mathbb{L}_2(\mathbb{R}^2)$ be a two dimensional image, $G = \mathrm{SE}(2)$ denotes the roto-translation group then*

$$\rho(g)\Phi(\boldsymbol{y}) = \Phi(\boldsymbol{R}_\theta^{-1}(\boldsymbol{y} - \boldsymbol{x})).$$

We continue this part with some additional definitions needed in the next part.

**Definition 6** (Coset). *Let $H \subset G$ be a subgroup of $G$. Then $gH$ denotes a coset given by*

$$gH = \left\{g \cdot h \mid h \in H\right\}.$$

**Definition 7** (Quotient Space). *Let $H \subset G$ be a subgroup of $G$. Then $G/H$ denotes the quotient space that is defined as the collection of unique cosets $gH \subset G$. Elements of $G/H$ are thus cosets that represents an equivalence class of transformations for which $g \sim \tilde{g}$ are equivalent if there exists an $h \in H$ such that $g = \tilde{g}h$.*

**Definition 8** (Stabilizer). *Let $G$ act on $\mathcal{X}$ via the action $\odot$. For every $\boldsymbol{x} \in \mathcal{X}$, the stabilizer subgroup of $G$ with respect to the point $\boldsymbol{x}$, denoted as $\mathrm{Stab}_G(\boldsymbol{x})$, is the set of all elements in $G$ that fix $\boldsymbol{x}$, i.e.*

$$\mathrm{Stab}_G(\boldsymbol{x}) = \left\{g \in G \mid g \odot \boldsymbol{x} = \boldsymbol{x}\right\}.$$

Moreover from Bekkers (2019) we know that, if $\mathcal{X}$ is a homogeneous space (informally, a homogeneous space can be understood as a space that possesses a uniform structure throughout, in the sense that every point can be transformed into any other by the action of a group.) of $G$, then $\mathcal{X}$ can be identified with $G/H$ with $H = \mathrm{Stab}_G(\boldsymbol{x}_0)$ for any $\boldsymbol{x}_0 \in \mathcal{X}$. For simplicity in notation we do not use $\cdot$ and $\odot$ symbols in the next sections. Furthermore, in this paper we use $g$ and $h$ to denote group elements and $\Phi$ and $K$ to denote input and kernel functions respectively.

## 3 Group Convolutional Neural Networks Architecture

One conventional method to build group convolutional neural networks is to apply isotropic convolutions for Equation (1). An isotropic $\mathbb{R}^d$ convolutional layer maps between planar signals $\mathbb{L}_2\left(\mathbb{R}^d\right)$ with $\mathcal{K}$ a planar correlation given by

$$(\mathcal{K}\Phi)(\boldsymbol{y}) = \int_{\mathbb{R}^d} K(\boldsymbol{x} - \boldsymbol{y})\Phi(\boldsymbol{x})\mathrm{d}\boldsymbol{x},$$

and in which it is shown (Theorem 1 from Bekkers (2019)) that if $\mathcal{Y} \equiv G/H$ is the quotient of $G$ with $H = \mathrm{Stab}_G(y_0) = \{g \in G \mid gy_0 = y_0\}$, then the kernel $K$ satisfies

$$\text{for all } h \in H : \quad K(\boldsymbol{x}) = \frac{1}{|\det h|} K\left(h^{-1}\boldsymbol{x}\right). \tag{2}$$

Applying isotropic convolutions is limiting because they are constrained by the shape of the kernels. One approach to overcome this limitation, is to lift the signals to a group $G$. Lifting of the input signal, not only addresses the constraints of kernels as noted by Bekkers (2019) but also offers advantages in enhancing the

performance of image processing, as highlighted in the work by Smets et al. (2023). When we apply lifting we must look for stabilizer $\text{Stab}_G$ when $G$ acts on $G$. In this case we have

$$H = \text{Stab}_G(g) = \{x \in G | xg = g\} = \{e\}.$$

As a result, Equation (2) is fulfilled for all kernels, and there are no longer any limitations imposed on the choice of kernels.

**Definition 9** (Lifting layer ($\mathcal{X} = \mathbb{R}^d, \mathcal{Y} = G_2$)). *Let* $g = (\boldsymbol{x}, \boldsymbol{P}) \in G_2$, *where* $\boldsymbol{x} \in \mathbb{R}^2$ *and* $\boldsymbol{P} \in \text{GL}_2(\mathbb{R})$. *A lifting layer maps functions from* $\mathbb{L}_2(\mathbb{R}^d)$ *to* $\mathbb{L}_2(G_2)$, *where* $G_2$ *is a group, via a lifting correlation defined by:*

$$(\mathcal{K}f)(g) = \int_{\mathbb{R}^d} \frac{1}{|\det \boldsymbol{P}|} K\left(g^{-1}\tilde{\boldsymbol{x}}\right) \Phi(\tilde{\boldsymbol{x}}) \mathrm{d}\tilde{\boldsymbol{x}}.$$

**Example 4** (Lifting for Kronecker delta kernel). *Let*

$$K = \delta(\boldsymbol{x}, \boldsymbol{0}_{d\times d}) = \begin{cases} 1 & \text{if } \boldsymbol{x} = \boldsymbol{0} \in \mathbb{R}^d; \\ 0 & \text{otherwise.} \end{cases}$$

*Then for* $g = (\boldsymbol{x}, \boldsymbol{P})$ *we have*

$$g^{-1}\tilde{\boldsymbol{x}} = \boldsymbol{P}^{-1}(\tilde{\boldsymbol{x}} - \boldsymbol{x}).$$

*Therefore,*

$$K(g^{-1}\tilde{\boldsymbol{x}}) = \delta(\boldsymbol{P}^{-1}(\tilde{\boldsymbol{x}} - \boldsymbol{x}), \boldsymbol{0}_{d\times d}) = \begin{cases} 1 & \text{if } \tilde{\boldsymbol{x}} = \boldsymbol{x}; \\ 0 & \text{otherwise.} \end{cases}$$

*This implies that the lifting layer is given by*

$$(\mathcal{K}\Phi)(g) = \frac{\Phi(\boldsymbol{x})}{|\det \boldsymbol{P}|}.$$

We also need to discuss the existence of the lifting layer integral. A function $\Phi$ on $\mathbb{R}$ is called locally integrable if $\Phi$ is integrable on every bounded interval $[a, b]$ for $a < b$ in $\mathbb{R}$. If $K \in \mathcal{C}_c^\infty(\mathbb{R})$ and $\Phi$ is locally integrable, then

$$(\Phi * K)(y) = \int_{-\infty}^{\infty} \Phi(t)K(y - t)\mathrm{d}t,$$

exists and is infinitely differentiable on $\mathbb{R}$. First of all the input $\Phi$ is usually a picture and therefore the function $\Phi$ is bounded. On the other hand the value of lifted functions on cosets is equal to that of $\Phi$. Therefore the lifted function is bounded as well. We further know that the kernel is locally supported, which results the integrability. After lifting layer we will apply convolutional layer which is defined as follows.

**Definition 10** (Group convolutional layer ($\mathcal{X} = \mathcal{Y} = G$)). *A group convolutional layer maps between* $G$-*feature maps in* $\mathbb{L}_2(G)$. *A group convolution is given by*

$$(\Phi * K)(h) = \int_G \Phi(h)K\left(h^{-1}g\right) \mathrm{d}\mu_G,$$

*where* $g \in G$ *and* $\mu_G$ *is a Haar measure.*

We also need another layer to again map to feature maps in $\mathbb{L}_2(\mathbb{R}^d)$. Let $G = \mathbb{R}^d \rtimes \tilde{H}$. Then we have the following definition:

**Definition 11** ($\mathbb{R}^d$ Projection layer). *A projection layer maps between* $G$-*feature maps in* $\mathbb{L}_2(G)$ *back to planar feature maps in* $\mathbb{L}_2(\mathbb{R}^d)$

$$(\mathcal{K}\Psi)(\boldsymbol{x}) = \int_{\tilde{H}} \Psi(\boldsymbol{x}, \tilde{h}) \mathrm{d}\tilde{h}. \tag{3}$$

Frequently, the focus is on constructing architectures that are invariant, as opposed to equivariant. Invariance with respect to all transformations in $G$ is accomplished through mean pooling across the entire group $G$, akin to how global translation invariance is typically obtained by mean or max pooling over the spatial dimensions of feature maps. The global pooling layer can be defined as follows.

**Definition 12** (Global pooling layer). *A global pooling layer transforms any feature map into a single scalar value as follows,*

$$(\mathcal{K}\Phi) = \int_{\mathcal{X}} \Phi(\boldsymbol{x})\mathrm{d}\mu(\boldsymbol{x}), \tag{4}$$

*where $\mathcal{K}$ represents a pooling operation over $\mathcal{X}$ and $d\mu(\boldsymbol{x})$ is a Radon measure on $\mathcal{X}$.*

# 4 Principal Results

Here, we provide the principal result established in this section. In particular we explore affine invariant spaces and investigate the convolution integration over $G_2$.

## 4.1 Problem Statement

Our goal is to study invariance in affine transformations in continuous-domain convolutional neural networks. An affine transformation basically combines linear transformations and translations. Affine transformations are denoted as follows

$$G_2 = \left\{ [\boldsymbol{x}, \boldsymbol{A}] : \boldsymbol{x} \in \mathbb{R}^2, \boldsymbol{A} \in \mathrm{GL}_2(\mathbb{R}) \right\},$$

where

$$[\boldsymbol{x}, \boldsymbol{A}] : \boldsymbol{z} \mapsto \boldsymbol{x} + \boldsymbol{A}\boldsymbol{z}.$$

The identity is $[\boldsymbol{0}, \boldsymbol{I}]$, and, therefore, for all $\boldsymbol{B} \in \mathrm{GL}_2(\mathbb{R})$ we have $[\boldsymbol{y}, \boldsymbol{B}]^{-1} = [-\boldsymbol{B}^{-1}\boldsymbol{y}, \boldsymbol{B}^{-1}]$.

The affine transformation is important as we may face affine type distortions due to varying proximity of the camera with respect to the object. For example, this type of affine distortion could manifest in remote sensing images, as well as in camera imagery which can include various perspective distortions Fisher et al. (2000). It is important to note that in an affine transformation, parallel lines in the original image continue to remain parallel in the transformed image. However, the transformation can introduce distortion in the angles between lines. In order to study resemblance of two input functions $\Phi_1$ and $\Phi_2$ we need to provide some more definitions.

**Definition 13** ($\epsilon$-equivalance). *We say that functions $\Phi_1, \Phi_2 \in \mathbb{L}(\mathbb{R}^2)$ are $\epsilon$-equivalent if $\|\Phi_1 - \Phi_2\|_1 < \epsilon$ or $\sup_{\boldsymbol{x}} |\Phi_1(\boldsymbol{x}) - \Phi_2(\boldsymbol{x})| < \epsilon$.*

For instance, if a picture shows a slight deviation due to noise, we aim to overlook or disregard this deviation.

**Definition 14** (Affine invariance). *We say that functions $\Phi_1, \Phi_2 \in \mathbb{L}(\mathbb{R}^2)$ are Affine invariant if there exists $h \in G_2$ so that $\Phi_1 = \rho(h)\Phi_2$.*

**Definition 15** ($\epsilon$-Affine invariance). *We say that functions $\Phi_1, \Phi_2 \in \mathbb{L}(\mathbb{R}^2)$ are $\epsilon$-Affine invariant if there exists $h \in G_2$ so that $\|\Phi_1 - \rho(h)\Phi_2\|_1 < \epsilon$ or $\sup_{\boldsymbol{x}} |\Phi_1(\boldsymbol{x}) - \rho(h)\Phi_2(\boldsymbol{x})| < \epsilon$, where $\rho$ denotes a representation.*

This paper explores the use of convolutional neural networks in handling affine transformations, focusing specifically on cases where the transformation matrix $\boldsymbol{A}$ belongs to the general linear group $\mathrm{GL}_2(\mathbb{R})$. We diverge from the use of isometric convolutions, opting instead for the application of the lifting-projection method, which we have elucidated in Section 3 comprehensively. While prior investigations have focused on compact groups such as $\mathrm{SO}(2)$, it is important to highlight that the $\mathrm{GL}_2(\mathbb{R})$ group does not fall under the category of compact groups. Our alternative method focuses on analyzing the convolution of the lifted forms of the signals $\Phi_1$ and $\Phi_2$ for achieving $G_2$ invariance. We also need to introduce an extra step that encompasses performing convolutions on $G_2$ and address the unique challenges associated with this, including techniques for handling integrations over $G_2$.

Therefore, our first goal is to establish the stability of the proposed three-layer group-convolutional neural network under affine transformations generated by the general linear group $\mathrm{GL}_2(\mathbb{R})$. For clarity, we first state the main stability result, which summarizes the behavior of the overall construction. The subsequent definitions and examples are then introduced to provide intuition and to make the construction explicit.

**Theorem 1** (Stability of G-CNN architecture). *Let $\Sigma$ be the G-CNN consisting of three layers: lifting, convolutional, and $\mathbb{R}$-projection and let the distance of a function $\Phi_1$ and the affine transform of another function $\Phi_2$ be less than $\epsilon$, then*

$$|\Sigma\Phi_1 - \Sigma\Phi_2| < c\epsilon,$$

*where $c = \|K_1\|_1^{\mathbb{R}^2}\|K_2\|_1^{G_2}$ and $K_1$ and $K_2$ are the kernels of the lifting layer and the convolutional layer, respectively. Where $\|K\|_1^{\mathbb{R}^2} := \int_{\mathbb{R}^2} |K(\boldsymbol{x})| \, d\boldsymbol{x}$ and $\|K_2\|_1^{G_2} = \int_{G_2} |K_2| \, d\mu_{G_2}$.*

*Proof.* We analyze the three layers of the network individually, demonstrating the stability of each layer. This, in turn, ensures the stability of the entire network. Theorems 2, 3, and 4 detail this process. The current theorem is a direct consequence of these theorems. □

**Remark 1.** *Example 4 implies that the stability of affine-invariant systems can decrease as the determinant of the transformation matrix approaches zero. This aligns with geometric intuition: as $\det(\boldsymbol{P}) \to 0$, the transformation projects initial objects up to a very small distance to lower-dimensional subspace. Consequently, inversion or reconstruction becomes more challenging.*

The next theorem asserts that, the lifting layer does not change the affine invariance of input signals. The proof of the following theorems can be found in Section A.

**Theorem 2** (Invariance in the lifting layer). *Let $\Phi_1, \Phi_2 : \mathbb{R}^2 \to \mathbb{R}$ be input signals. Suppose there exists $h \in G_2$ such that*

$$\sup_{\boldsymbol{x}\in\mathbb{R}^2} \left|\Phi_1(\boldsymbol{x}) - \rho(h^{-1})\Phi_2(\boldsymbol{x})\right| < \epsilon.$$

*Then*

$$\sup_{g\in G_2} \left|(K\Phi_1)(g) - \rho(h^{-1})(K\Phi_2)(g)\right| < \epsilon\|K\|_1^{\mathbb{R}^2}.$$

The next step is to demonstrate the invariance in the convolutional layer.

**Theorem 3** (Invariance in the convolutional layer). *Let $(\mathcal{K}\Phi_1), (\mathcal{K}\Phi_2) : G_2 \to \mathbb{R}$ be the lifting of $\Phi_1, \Phi_2 : \mathbb{R}^2 \to \mathbb{R}$. If there exists an $\tilde{h} \in G_2$ so that $\|(\mathcal{K}\Phi_1)(g) - \rho(\tilde{h})(\mathcal{K}\Phi_2)(g)\|_{\sup} < \epsilon$ then*

$$\|(\mathcal{K}\Phi_1) * K - \rho(\tilde{h})(\mathcal{K}\Phi_2) * K\|_{\sup}^{G_2} < \epsilon\|K\|_1^{G_2}$$

*holds for every kernel $K$ and vice-versa. Here, $\|\Phi\|_1^{G_2} = \int_{G_2} |\Phi| \, d\mu_{G_2}$ denotes the norm induced by integration over $G_2$, and $*$ denotes the convolution operator.*

The aforementioned finding indicates that to assess the equivalence of two signals, it is necessary to perform a convolutional integration across $G_2$. We investigate this problem in the next section. In the last step we provide the below theorem which states that the function defined by $\int_{G_2}(\mathcal{K}\Phi) * K \, d\mu_{G_2}(g)$ which is a continuous function from $\mathcal{C}(\mathbb{R}^2, \mathbb{R})$ to $\mathbb{R}$ can be used for characterization of invariant affine functions in the projection and pooling layer.

**Theorem 4** (Invariance in the projection and pooling layer). *If $(\mathcal{K}\Phi_1), (\mathcal{K}\Phi_2) : G_2 \to \mathbb{R}$ are lifting of input signals and there exists a $\tilde{h} \in G_2$ such that $\|(\mathcal{K}\Phi_1) - \rho(\tilde{h})(\mathcal{K}\Phi_2)\|_1^{G_2} < \epsilon$. Then we have*

$$\left|\int_{G_2} \left((\mathcal{K}\Phi_1) * K - (\mathcal{K}\Phi_2) * K\right)(h) d\mu_{G_2}(h)\right| \leq \epsilon\|K\|_1^{G_2}$$

## 4.2 Convolution Computation

Before illustrating how to compute the convolution over the group $G_2$, we remark some ingredients which are essential to compute the convolution over $G_2$. We finally show that the convolution over $G_2$ can be computed through Fourier transform and integration over real valued space.

In our study, we adopt a straightforward approach to calculate the $G_2$-invariant convolution for a kernel, which can be formulated as follows:

$$\int_{G_2} \Phi([\boldsymbol{x}, \boldsymbol{A}])K([\boldsymbol{y}, \boldsymbol{B}]^{-1}[\boldsymbol{x}, \boldsymbol{A}])d\mu_{G_2}. \tag{5}$$

Using the Stone–Weierstrass theorem, in the setup of continuous functions with respect to sup-norm, $\mathcal{C}(G_2, \mathbb{R}) = \mathcal{C}(\mathrm{GL}_2(\mathbb{R}) \ltimes \mathbb{R}^2, \mathbb{R})$, which asserts that sums of separable functions are dense in $\mathcal{C}(G_2, \mathbb{R})$, we reduce the kernel sets to functions of the form $K(\boldsymbol{y}, \boldsymbol{A}) = \sum_{i=1}^{M} K_{1_i}(\boldsymbol{y}) K_{2_i}(\boldsymbol{A})$. This reduction helps us to benefit from Fourier transforms to simplify some parts of our calculations. We use QR parametrization of $\mathrm{GL}_2(\mathbb{R})$ which aids us in utilizing numerical approaches, for example those introduced in Eshkuvatov et al. (2013). Now, we illustrate the outcomes presented in Milad & Taylor (2023); Schindler (1993), which are pertinent to our calculations. Let

$$K_0 = \left\{ \begin{pmatrix} s & -t \\ t & s \end{pmatrix} : s, t \in \mathbb{R}, s^2 + t^2 > 0 \right\},$$

and

$$H_{(1,0)} = \left\{ \begin{pmatrix} 1 & 0 \\ u & v \end{pmatrix} : u, v \in \mathbb{R}, v \neq 0 \right\}.$$

It is shown that $\mathrm{GL}_2(\mathbb{R}) = K_0 H_{(1,0)}, K_0 \cap H_{(1,0)} = \boldsymbol{I}$, where $\boldsymbol{I}$ denotes the identity matrix, and $(\boldsymbol{M}, \boldsymbol{C}) \to \boldsymbol{MC}$ is a homeomorphism between $K_0 \times H_{(1,0)}$ and $\mathrm{GL}_2(\mathbb{R})$. From Milad & Taylor (2023) we have

$$\int_{G_n} \Phi \, \mathrm{d}\mu_{G_n} = \int_{\mathrm{GL}_n(\mathbb{R})} \int_{\mathbb{R}^n} \Phi[\boldsymbol{x}, \boldsymbol{A}] \frac{\mathrm{d}\boldsymbol{x} \mathrm{d}\mu_{\mathrm{GL}_n(\mathbb{R})}(\boldsymbol{A})}{|\det(\boldsymbol{A})|}, \text{ for all } \Phi \in C_c(G_n), \tag{6}$$

where $\mathcal{C}_c(G_n)$ denotes the space of continuous $\mathbb{R}$-valued functions of compact support on $G_n$. For any integrable function $\Phi$ on $\mathrm{GL}_2(\mathbb{R})$, the Haar integral on $\mathrm{GL}_2(\mathbb{R})$ can be expressed as

$$\int_{\mathrm{GL}_2(\mathbb{R})} \Phi \, \mathrm{d}\mu_{\mathrm{GL}_2(\mathbb{R})} = \int_{K_0} \int_{H_{(1,0)}} \Phi(\boldsymbol{MC}) |\det(\boldsymbol{C})| \mathrm{d}\mu_{H_{(1,0)}} \mathrm{d}\mu_{K_0}. \tag{7}$$

Let $\mathbb{R}^* = \mathbb{R} \setminus \{0\}$. Then the map

$$[u, v] \to \begin{pmatrix} 1 & 0 \\ u & v \end{pmatrix}$$

is an isomorphism between the group $G_1$ and $H_{(1,0)}$. When $n = 1, \mathrm{GL}_1(\mathbb{R})$ can be identified with $\mathbb{R}^*$ and $G_1$ can be identified with $\mathbb{R} \rtimes \mathbb{R}^*$. We recall that $\int_{\mathbb{R}^*} \Phi \mathrm{d}\mu_{\mathbb{R}^*} = \int_{\mathbb{R}} \Phi(b) \frac{\mathrm{d}b}{|b|}$, where the integral on the right hand side is the Lebesgue integral on $\mathbb{R}$, and by 6

$$\int_{G_1} \Phi \, \mathrm{d}\mu_{G_1} = \int_{\mathbb{R}} \int_{\mathbb{R}} \Phi[y, b] \frac{\mathrm{d}y \mathrm{d}b}{b^2}. \tag{8}$$

### 4.3 Integral over $G_2$

A difficult aspect in the implementation of group convolutional neural networks involves performing convolutions across the group. This segment addresses this particular challenge by delving into the problem, which we will break down into the more manageable tasks of calculating Fourier transforms and conducting integrations in real-valued space. We have the below theorem for the integration over $G_2$.

**Theorem 5.** *Let*

$$\boldsymbol{A} = \begin{pmatrix} a & b \\ c & d \end{pmatrix} \in \mathrm{GL}_2(\mathbb{R})$$

*and let the kernel be separable meaning that $K(\boldsymbol{x}, \boldsymbol{A}) = K_1(\boldsymbol{x}) K_2(\boldsymbol{A})$ and consider the one to one transform between $H$ and $H^*$ so that*

$$H^*(s, t, u, v, \boldsymbol{B}, \boldsymbol{y}) := H_{\Phi, K}(a, b, c, d, \boldsymbol{B}, \boldsymbol{y}) = \frac{K_2(\boldsymbol{A} \boldsymbol{B}^{-1})}{|\det(\boldsymbol{A})||\det(\boldsymbol{B}^{-1})|} \mathcal{F}^{-1} \left( \widehat{\Phi}(\boldsymbol{u}) \widehat{K}_1(\boldsymbol{B}^{\top} \boldsymbol{u}) \right),$$

*where $a = s - ut$, $c = t + us$, $b = -t/v$, and $d = s/v$, then we have*

$$\int_{G_2} \Phi([\boldsymbol{x}, \boldsymbol{A}]) K([\boldsymbol{y}, \boldsymbol{B}]^{-1}[\boldsymbol{x}, \boldsymbol{A}]) \mathrm{d}\mu_{G_2} = \int_{\mathbb{R}} \int_{\mathbb{R}} \int_{\mathbb{R}} \int_{\mathbb{R}} H^*(s, t, u, v, \boldsymbol{B}, \boldsymbol{y}) \frac{\mathrm{d}u \mathrm{d}v}{|v|} \frac{\mathrm{d}s \mathrm{d}t}{s^2 + t^2}.$$

*where $\mathcal{F}(K_1) = \widehat{K}_1(\boldsymbol{u})$ and $\mathcal{F}(\Phi) = \widehat{\Phi}(\boldsymbol{u})$ are Fourier transforms.*

The proof of this theorem is discussed in Section A. Applying this result we can use the numerical methods in Eshkuvatov et al. (2013) to compute the former integral as it has singularity in $s = 0, t = 0$. Note that we can write $K_0$ as $\mathbb{R}^+ \rtimes \mathrm{SO}(1)$ where

$$\begin{pmatrix} s & -t \\ t & s \end{pmatrix} = (s^2 + t^2) \times \begin{pmatrix} r \cos \theta & -r \sin \theta \\ r \sin \theta & r \cos \theta \end{pmatrix}.$$

The final step that necessitates computation is the integration within the projection layer. In the context of our affine transformation, the stabilizer is specifically $\mathrm{GL}_2(\mathbb{R})$. We refrain from delving into the intricacies of this process, as it bears resemblance to the earlier scenario.

Now we first provide one example on $G_1$ this example gives us some insights for examples on $G_2$. We also will see how two class of complicated functions are equivalent with some intervals on $\mathbb{R}$ which are easier to investigate.

**Example 5.** *Let the input function be defined as*

$$\Phi(a) = \begin{cases} 1 & if\ a \in [t_1, t_2] \\ 0 & otherwise \end{cases},$$

*then according to Example (4) for the lifting of $\Phi$ we have*

$$(\mathcal{K}\Phi)[a, b] = \begin{cases} \frac{1}{|b|} & if\ a \in [t_1, t_2] \\ 0 & otherwise \end{cases},$$

*then we have:*

$$\int_{\mathbb{R}^2} \mathcal{K}(\Phi[a, b]) K([x, y]^{-1}[a, b]) \mathrm{d}\mu_{G_1}(a, b)$$

$$= \int_{\mathbb{R}^2} \frac{1}{|b|} K([-\frac{y}{x}, \frac{1}{x}][a, b]) \frac{\mathrm{d}a\mathrm{d}b}{b^2}$$

$$= \int_{\mathbb{R}^2} \frac{1}{|b|b^2} K([\frac{a-y}{x}, \frac{b}{x}]) \mathrm{d}a\mathrm{d}b$$

*We can define a separable kernel so that we make the computations easier. We define it as $K(s, t) = s^3 \exp(s) \frac{1}{\sqrt{2\pi}} exp(-t^2)$. Employing this definition and assuming $b$ is positive, we obtain:*

$$\int_{\mathbb{R}^2} \frac{1}{|b|b^2} K([\frac{a-y}{x}, \frac{b}{x}]) \mathrm{d}a\mathrm{d}b = \int_{t_1}^{t_2} \exp(\frac{a-y}{x}) \int_{\mathbb{R}} \frac{1}{|b|b^2} \frac{b^3}{x^3} \frac{1}{\sqrt{2\pi}} \exp{-(\frac{b}{x})^2} \mathrm{d}a\mathrm{d}b$$

$$= \int_{t_1}^{t_2} \exp(\frac{a-y}{x}) \int_{\mathbb{R}} \frac{1}{|b|b^2} \frac{b^3}{x^3} \frac{1}{\sqrt{2\pi}} \exp{-(\frac{b}{x})^2} \mathrm{d}a\mathrm{d}b = \frac{1}{\sqrt{2}x} \exp(\frac{t_2-y}{x}) - \frac{1}{\sqrt{2}x} \exp(\frac{t_1-y}{x})$$

Figure 1 visualizes the outcome of Example 5. It shows that functions defined on different intervals, when related by an affine transformation, produce similar convolution outputs. This illustrates that affine invariance of more complex signals can be understood through simple step functions, and highlights the role of the kernel in preserving this relationship.

Figure 1 indeed illustrates how the proposed framework captures affine invariance using simple functions. The purpose of this example is not to restrict the analysis to simple signals, but rather to demonstrate that the affine invariance of more complex shapes and curves can be understood through their decomposition into simpler components. In this sense, the figure provides a conceptual bridge between the theoretical formulation and practical signals, showing how the invariance properties extend naturally from simple functions to general geometric structures.

To provide an intuitive understanding of the proposed construction, we summarize the affine-equivariant layer in simple terms. Given an input image, we first generate multiple transformed versions by applying a set

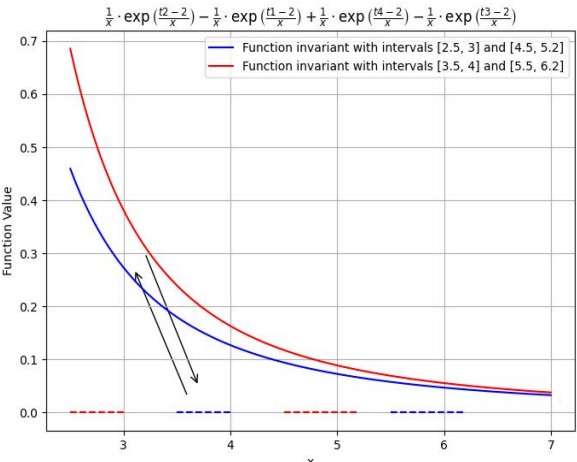

Figure 1: Two families of related affine invariant functions denoted by dashed and solid lines.

of affine transformations. Each of these transformed images is then processed using the same convolutional filters. Finally, the resulting feature maps are combined by averaging across all transformations. This process can be viewed as observing the input from different geometric perspectives and extracting features that remain consistent across these views. By aggregating these responses, the model produces representations that are robust to affine transformations. This procedure can be implemented using standard convolution operations applied to transformed inputs, followed by averaging.

## 5 Experiments

We evaluate the proposed method on both synthetic and real-world datasets to assess its robustness to affine transformations. The experiments are designed to provide controlled evaluation under known transformations and to validate performance on benchmark datasets.

A key challenge in implementing group convolution over the continuous affine group $G_2 = \mathbb{R}^2 \rtimes \mathrm{GL}_2(\mathbb{R})$ is that it requires integration over a non-compact group. In practice, such integrals are intractable. Therefore, we approximate group convolution using a finite set of sampled affine transformations.

Specifically, given a finite set $\{g_i\}_{i=1}^{|G|} \subset G_2$, the group convolution

$$(\Phi * K)(h) = \int_{G_2} \Phi(g)K(h^{-1}g)\,d\mu_{G_2}(g)$$

is approximated by

$$(\Phi * K)(h) \approx \frac{1}{|G|} \sum_{i=1}^{|G|} \Phi(g_i)K(h^{-1}g_i),$$

which corresponds to a discrete (Monte Carlo) approximation of the Haar integral over the affine group. This approximation is used throughout all experiments.

For the synthetic dataset, we additionally construct transformations based on Example 5 to provide a controlled setting aligned with the theoretical analysis.

For clarity, we summarize the practical implementation of the proposed affine-equivariant layer used in our experiments. Given an input image, a finite set of affine transformations $\{g_i\}_{i=1}^{|G|}$ is applied to produce transformed copies of the input (lifting stage).

Each transformed image is then processed independently using a fixed bank of $K = 7$ convolution kernels of size $3 \times 3$. This corresponds to applying shared convolutional operators across sampled group elements, approximating evaluation of the group convolution kernel.

To obtain a transformation-invariant representation, the resulting feature maps are averaged over the transformation dimension (projection stage). This averaging operation provides a discrete approximation of integration over the affine group.

In our implementation, the affine transformations, convolution kernels, and projection operator are all fixed. Consequently, the only trainable parameters are those of the final fully connected classifier, which is trained using standard backpropagation with cross-entropy loss.

### 5.1 Implementation Details

All input images are resized or zero-padded to $40 \times 40$ pixels and normalized to the range $[0, 1]$. The proposed GCNN architecture consists of three stages: lifting, affine group convolution (discretely approximated), and projection, followed by pooling and classification.

In the lifting stage, each input image is transformed using a finite set of affine transformations constructed from scaling, shearing, and translation. In total, $|G| = 15$ transformations are used. Translations are sampled uniformly within a range of $\pm 5$ pixels in both spatial dimensions. Each transformation is applied using bilinear interpolation with zero padding. To ensure consistency across transformations, each transformed image is normalized by the absolute determinant of the corresponding affine matrix.

**Lifting for general signals.** The lifting operator is defined via affine transformations of the input signal, i.e., $\mathcal{L}f(g, x) = f(g^{-1} \cdot x)$, rather than through an explicit kernel function. For general signals, this is implemented using interpolation (bilinear interpolation in our experiments), ensuring applicability beyond discrete or piecewise-constant inputs.

The lifting operator is implemented directly via affine transformations, providing a practical discretization applicable to general image signals.

The lifted representations are processed by an affine group convolution layer. A fixed bank of $K = 7$ convolution kernels of size $3 \times 3$ is applied independently to each transformed image using sliding-window operations. This procedure can be interpreted as evaluating the convolution kernel over a discretized subset of the affine group.

A nonlinear mapping

$$f \leftarrow \text{sign}(f) \left(1 - \exp(-|f|/\alpha)\right)$$

is applied to stabilize responses across transformations.

Following convolution, a projection step aggregates features across the transformation dimension by averaging. This produces a representation that is approximately invariant to the sampled affine group while preserving spatial structure.

The resulting feature maps are passed through a ReLU activation and a $2 \times 2$ average pooling operation, reducing the spatial resolution from $38 \times 38$ to $19 \times 19$. The final feature representation consists of 7 channels of size $19 \times 19$, yielding a feature vector of dimension 2527.

For classification, we use a fully connected neural network with two hidden layers of sizes 256 and 128, each followed by ReLU activation and dropout (rate 0.25). The output layer produces class probabilities over 10 classes.

The lifting and projection operators are fixed and not learned. The convolution kernels are also fixed. Consequently, the only trainable parameters are those of the fully connected classifier.

All models are trained using the Adam optimizer with a learning rate of $10^{-3}$ for 35 epochs and a batch size of 128. Cross-entropy loss is used as the training objective. Continuous group convolution is approximated by averaging over the finite set of sampled affine transformations.

**Trainable parameters.** We explicitly distinguish between fixed and learnable components of the proposed architecture. The lifting operator is implemented via a predefined finite set of affine transformations and is not learned. Similarly, the projection operator is implemented as averaging over the transformation dimension and is also fixed.

In the current implementation, the convolution kernels used in the affine group convolution layer are fixed and shared across all transformed inputs. As a result, the only trainable parameters in the model are those of the final fully connected classifier, which is optimized using standard backpropagation with cross-entropy loss.

This design choice allows us to isolate the effect of affine-aware representations independently of kernel learning. We note that the framework naturally supports learnable convolution kernels in the group domain, which we leave as future work.

### 5.2 Synthetic Affine Transformation Experiments

We first evaluate the proposed method on a controlled synthetic dataset to isolate the effect of affine transformations.

Digit images (0–9) are rendered on a fixed grayscale canvas and transformed using affine transformations incorporating stretch, shear, rotation, and translation. For each digit class, $N$ samples are generated, resulting in a dataset of size $10N$. The dataset is split into training and test sets using an 80/20 split.

In the first experiment, we use the affine transformation

$$A_1 = \begin{bmatrix} 2.5 & 0.7 \\ 0.6 & 1.8 \end{bmatrix}.$$

As shown in Fig. 2, the GCNN outperforms the standard CNN across all dataset sizes. The GCNN demonstrates strong generalization even with limited data, while the CNN requires more samples to achieve comparable performance.

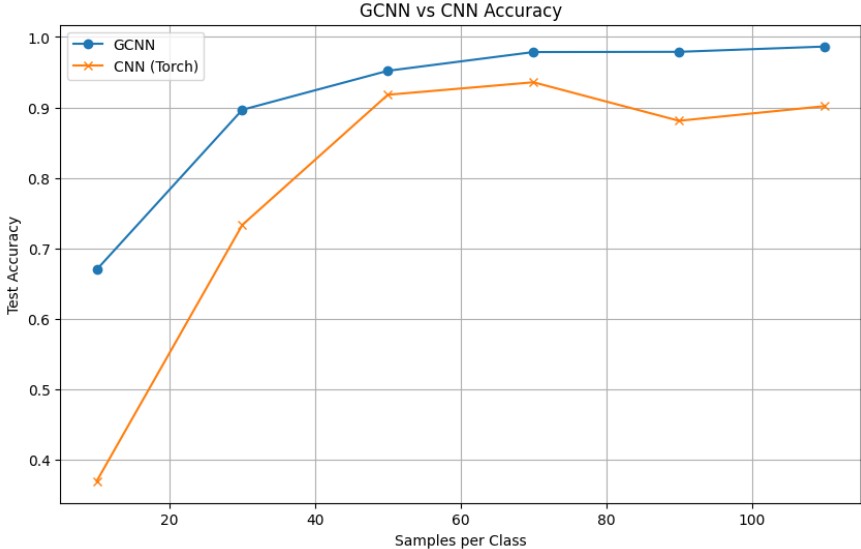

Figure 2: Accuracy comparison between GCNN and CNN under affine transformation $A_1$.

As the number of training samples increases, the performance gap between GCNN and CNN decreases. The GCNN encodes affine invariance explicitly, enabling strong generalization in low-data regimes. In contrast, the CNN must learn invariance from data and therefore requires more samples. With sufficient data, the CNN can approximate invariance and may match or exceed GCNN performance due to its greater flexibility.

We further evaluate using the transformation

$$A_2 = \begin{bmatrix} 1 & 0.7 \\ 0.7 & 1 \end{bmatrix}.$$

As illustrated in Fig. 3, the GCNN again outperforms the CNN when the input data size is limited. As the number of training samples increases, the performance gap between GCNN and CNN decreases, and CNN may outperform GCNN. This is because GCNN encodes affine invariance explicitly, which is advantageous in low-data regimes, while CNN benefits from larger datasets by learning invariance from data.

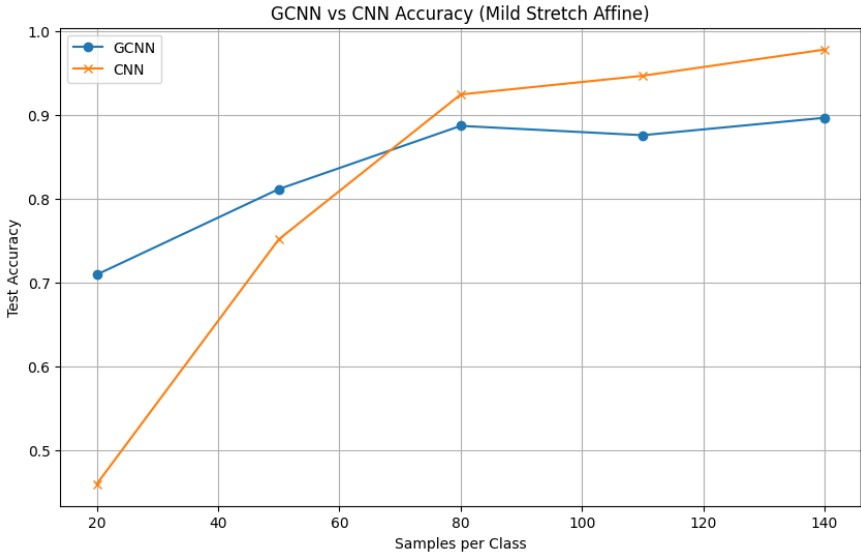

Figure 3: Accuracy comparison under affine transformation $A_2$.

**Qualitative Results.**  To further analyze model behavior, we consider additional affine transformations:

$$A_1 = \begin{bmatrix} 1 & 2 \\ 2 & 1 \end{bmatrix}, \quad A_2 = \begin{bmatrix} 1 & 0.7 \\ 0.7 & 1 \end{bmatrix}, \quad A_3 = \begin{bmatrix} 1 & 0.5 \\ 0.5 & 1 \end{bmatrix}.$$

Figures 4–6 present predictions for both models. The GCNN consistently exhibits stronger robustness to affine distortions, correctly identifying more transformed digits than the CNN baseline.

### 5.3   Evaluation on affNIST

We evaluate the proposed method on affNIST, which contains digits with strong affine distortions and provides a diverse range of transformations.

Figure 7 shows test accuracy over training epochs. The GCNN achieves a best test accuracy of **74.22%**, outperforming the CNN baseline (**69.57%**).

### 5.4   Evaluation on MNIST

To evaluate generalization, we test both models on MNIST. Images are zero-padded to $40 \times 40$.

Figure 8 shows test accuracy over training epochs. The GCNN achieves **93.70%**, while the CNN achieves **93.40%**. The performance of the two models is comparable due to the limited geometric variability in MNIST.

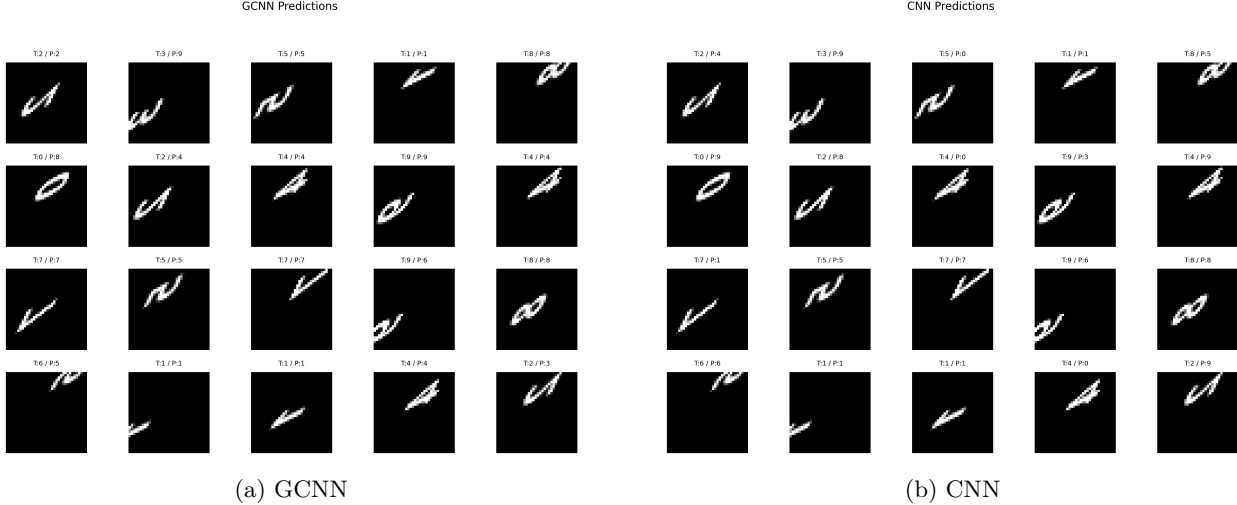

(a) GCNN        (b) CNN

Figure 4: Prediction comparison under affine transformation $A_1$.

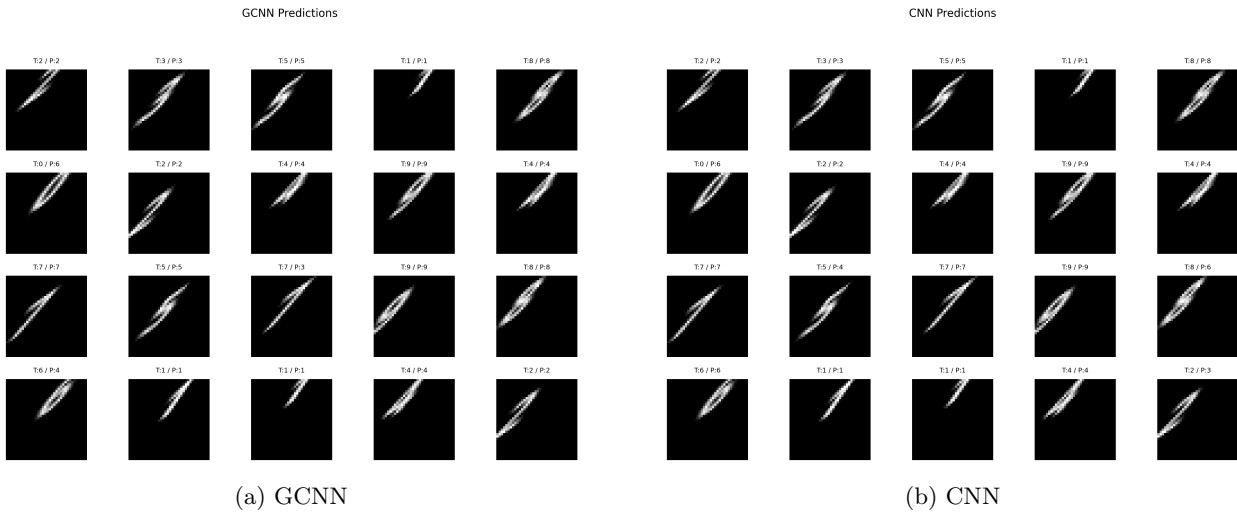

(a) GCNN        (b) CNN

Figure 5: Prediction comparison under affine transformation $A_2$.

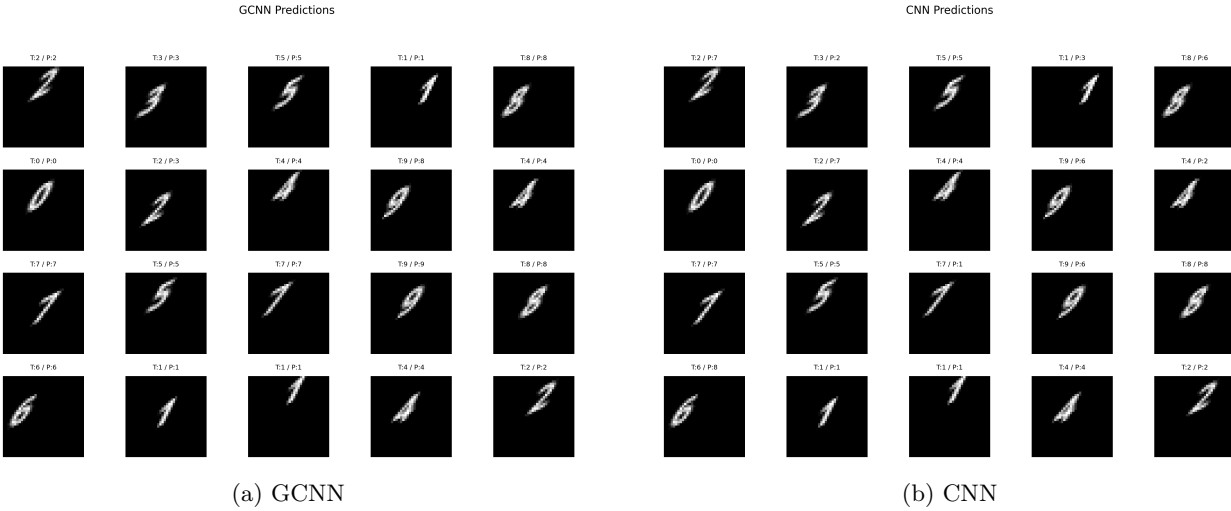

(a) GCNN            (b) CNN

Figure 6: Prediction comparison under affine transformation $A_3$.

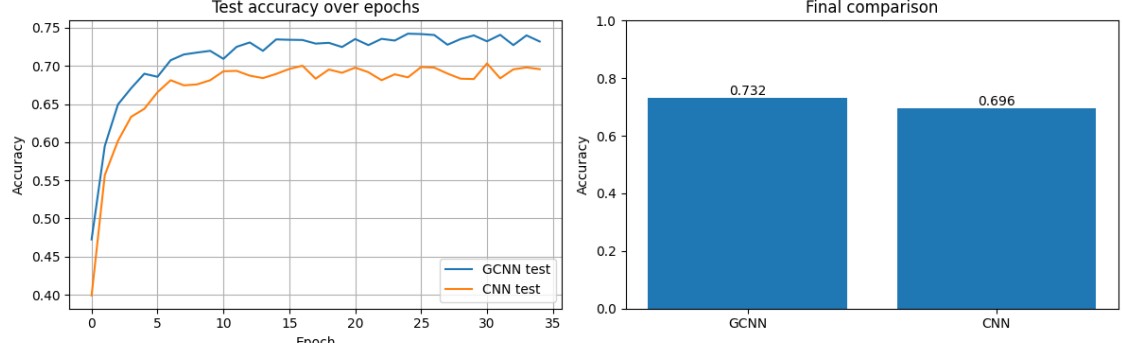

Figure 7: Test accuracy over epochs on affNIST.

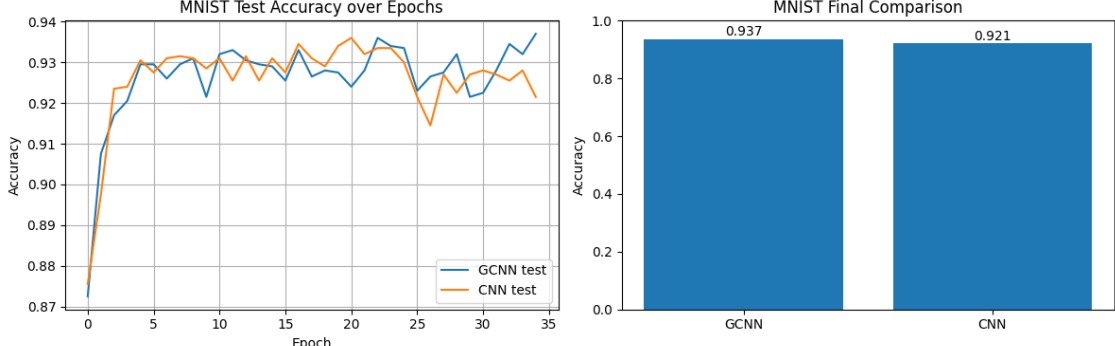

Figure 8: Test accuracy over epochs on MNIST.

The results indicate that the proposed model exhibits lower sensitivity to input perturbations compared to the CNN baseline, supporting the theoretical stability guarantees. Recent approaches such as MacDonald et al. (2022) approximate group convolutions via Monte Carlo integration over arbitrary Lie groups, while Li et al. (2024) construct affine-equivariant features using differential invariants. In contrast, our approach uses structured finite sampling combined with shared convolutional operators, yielding a simple and computationally efficient approximation that is straightforward to implement in practice.

A limitation of the current implementation is that convolution kernels are fixed and not learned. While this design isolates the effect of affine-aware representations, it may limit expressiveness compared to fully learnable architectures. However, the framework is not inherently restricted to fixed kernels, and extending it to learnable group convolution kernels is a natural direction for future work.

## 6 Conclusion

In this work, we investigated affine invariance in Group Convolutional Neural Networks (G-CNNs) under transformations generated by the general linear group $GL_2(\mathbb{R})$. We proposed a lifting–convolution–projection framework that provides a practical discrete approximation of affine-equivariant representations in the continuous domain.

Our analysis shows that the proposed architecture yields stable representations under affine transformations and that the associated computations can be reduced to tractable operations over $\mathbb{R}^2$. In addition to the theoretical formulation, we provided an explicit and implementable construction of the model, helping bridge the gap between abstract group convolution theory and practical neural network design.

Compared to existing approaches that rely on Monte Carlo integration over Lie groups or specialized invariant feature constructions, the proposed method offers a simple and computationally efficient alternative based on structured sampling and shared convolutional operators.

Experimental results on synthetic and benchmark datasets demonstrate improved robustness to affine distortions compared to a standard CNN baseline, particularly in low-data regimes. On datasets with limited geometric variability, such as MNIST, both models achieve comparable performance, highlighting the trade-off between built-in invariance and model flexibility.

**Limitations and Future Work.** The current implementation relies on fixed convolution kernels and a finite sampling of the affine group, which may limit expressiveness compared to fully learnable equivariant architectures. Additionally, the experimental evaluation is primarily conducted on digit datasets, and further validation on more complex data such as natural images or temporal signals is needed.

Future work will explore learnable group convolution kernels, more accurate approximations of continuous group integration, and applications to broader domains. A deeper empirical investigation of stability properties and their relationship to the theoretical bounds is also an important direction.

Overall, this work provides a practical and interpretable step toward incorporating affine structure into convolutional architectures, contributing to the development of models that are robust to geometric transformations.

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

## A  Technical Results and Proofs

In this section, we initially present an example analogous to Example 5 but within the context of $G_2$. Then we provide the proofs of the main theorems in the section.

**Example 6.** *Consider the below input function:*

$$\Phi(\boldsymbol{x}) = \begin{cases} 1 & if\ \boldsymbol{x} \in [t_1, t_2] \times [s_1, s_2] \\ 0 & otherwise \end{cases},$$

*then according to Example (4) for the lifting of $\Phi$ we have*

$$(\mathcal{K}\Phi)[\boldsymbol{x}, \boldsymbol{A}] = \begin{cases} \frac{1}{|\det(\boldsymbol{A})|} & if\ \boldsymbol{x} \in [t_1, t_2] \times [s_1, s_2] \\ 0 & otherwise \end{cases},$$

*as a result*

$$\begin{aligned}
&\int_{G_2} \Phi([\boldsymbol{x}, \boldsymbol{A}]) k([\boldsymbol{y}, \boldsymbol{B}]^{-1}[\boldsymbol{x}, \boldsymbol{A}]) \mathrm{d}\mu_{G_2} \\
&= \int_{G_2} \Phi([\boldsymbol{x}, \boldsymbol{A}]) k(\boldsymbol{B}^{-1}\boldsymbol{x} - \boldsymbol{B}^{-1}\boldsymbol{y}, \boldsymbol{A}\boldsymbol{B}^{-1}) \mathrm{d}\mu_{G_2} \\
&= \int_{G_2} \frac{1}{|\det(\boldsymbol{A})|} k(\boldsymbol{B}^{-1}\boldsymbol{x} - \boldsymbol{B}^{-1}\boldsymbol{y}, \boldsymbol{A}\boldsymbol{B}^{-1}) \mathrm{d}\mu_{G_2} \\
&= \int_{t_1}^{t_2} \int_{s_1}^{s_2} \int_{\mathrm{GL}_2} \frac{1}{|\det(\boldsymbol{A})|^2} k(\boldsymbol{B}^{-1}\boldsymbol{x} - \boldsymbol{B}^{-1}\boldsymbol{y}, \boldsymbol{A}\boldsymbol{B}^{-1}) \mathrm{d}\mu_{\mathrm{GL}_2} \mathrm{d}x_1 \mathrm{d}x_2,
\end{aligned}$$

*We define the separable kernel as follows*

$$k(\boldsymbol{x}, \boldsymbol{P}) = |\det(\boldsymbol{P})|^2 \exp(\langle \boldsymbol{x}, \boldsymbol{l} \rangle) \mathcal{N}(\boldsymbol{P}),$$

*where $\boldsymbol{l} = [l_1, l_2]$, and $\mathcal{N}(\boldsymbol{P})$ is the normalized heat kernel of $\mathrm{GL}_2(\mathbb{R})$. By this choice we obtain*

$$\begin{aligned}
&\int_{t_1}^{t_2} \int_{s_1}^{s_2} \int_{\mathrm{GL}_2(\mathbb{R})} \frac{1}{|\det(\boldsymbol{A})|^2} k(\boldsymbol{B}^{-1}\boldsymbol{x} - \boldsymbol{B}^{-1}\boldsymbol{y}, \boldsymbol{A}\boldsymbol{B}^{-1}) \mathrm{d}x_1 \mathrm{d}x_2 \mathrm{d}\mu_{\mathrm{GL}_2(\mathbb{R}} \\
&= \int_{t_1}^{t_2} \int_{s_1}^{s_2} \int_{\mathrm{GL}_2(\mathbb{R}} \frac{1}{|\det(\boldsymbol{A})|^2} |\det(\boldsymbol{B}^{-1})|^2 |\det(\boldsymbol{A})|^2 \exp(\langle \boldsymbol{B}^{-1}\boldsymbol{x} - \boldsymbol{B}^{-1}\boldsymbol{y}, \boldsymbol{l} \rangle) \mathcal{N}(\boldsymbol{P}) \mathrm{d}x_1 \mathrm{d}x_2 \mathrm{d}\mu_{\mathrm{GL}_2(\mathbb{R}} \\
&= \int_{t_1}^{t_2} \int_{s_1}^{s_2} \frac{1}{|\det(\boldsymbol{B})|^2} \exp(\langle \boldsymbol{B}^{-1}\boldsymbol{x} - \boldsymbol{B}^{-1}\boldsymbol{y}, \boldsymbol{l} \rangle) \int_{\mathrm{GL}_2(\mathbb{R}} \mathcal{N}(\boldsymbol{P}) \mathrm{d}\mu_{\mathrm{GL}_2(\mathbb{R}} \mathrm{d}x_1 \mathrm{d}x_2 \\
&= \frac{1}{|\det(\boldsymbol{B})|^2 \exp(\langle \boldsymbol{B}^{-1}\boldsymbol{y}, \boldsymbol{l} \rangle)} \int_{t_1}^{t_2} \int_{s_1}^{s_2} \exp(\langle \boldsymbol{B}^{-1}\boldsymbol{x}, \boldsymbol{l} \rangle) \mathrm{d}x_1 \mathrm{d}x_2
\end{aligned}$$

*Assume that $\boldsymbol{B}^{-1} = \begin{pmatrix} \beta_1 & \beta_2 \\ \beta_3 & \beta_4 \end{pmatrix}$, then we obtain*

$$\frac{1}{|\det(\boldsymbol{B})|^2 \exp(\langle \boldsymbol{B}^{-1}\boldsymbol{y},\boldsymbol{l}\rangle)} \int_{t_1}^{t_2} \int_{s_1}^{s_2} \exp(\langle \boldsymbol{B}^{-1}\boldsymbol{x},\boldsymbol{l}\rangle)\mathrm{d}x_1\mathrm{d}x_2$$

$$= \frac{1}{|\det(\boldsymbol{B})|^2 \exp(\langle \boldsymbol{B}^{-1}\boldsymbol{y},\boldsymbol{l}\rangle)} \int_{t_1}^{t_2} \int_{s_1}^{s_2} \exp(\langle \boldsymbol{B}^{-1}\boldsymbol{x},\boldsymbol{l}\rangle)\mathrm{d}x_1\mathrm{d}x_2$$

$$= \frac{1}{|\det(\boldsymbol{B})|^2 \exp(\langle \boldsymbol{B}^{-1}\boldsymbol{y},\boldsymbol{l}\rangle)} \int_{t_1}^{t_2} \int_{s_1}^{s_2} \exp(l_1\beta_1 x_1 + l_1\beta_2 x_2 + l_2\beta_3 x_1 + l_2\beta_4 x_2)\mathrm{d}x_1\mathrm{d}x_2$$

$$= \frac{1}{|\det(\boldsymbol{B})|^2 \exp(\langle \boldsymbol{B}^{-1}\boldsymbol{y},\boldsymbol{l}\rangle)} \int_{t_1}^{t_2} \int_{s_1}^{s_2} \exp\Big(x_1(l_1\beta_1 + l_2\beta_3) + x_2(l_1\beta_2 + l_2\beta_4)\Big)\mathrm{d}x_1\mathrm{d}x_2$$

$$= \frac{1}{|\det(\boldsymbol{B})|^2 \exp(\langle \boldsymbol{B}^{-1}\boldsymbol{y},\boldsymbol{l}\rangle)(l_1\beta_1 + l_2\beta_3)(l_1\beta_2 + l_2\beta_4)}\Big(\exp\Big(t_2(l_1\beta_1 + l_2\beta_3) - \exp\Big(t_1(l_1\beta_1 + l_2\beta_3)\Big)$$

$$\times \Big(\exp\Big(s_2(l_1\beta_2 + l_2\beta_4)\Big) - \exp\Big(s_1(l_1\beta_2 + l_2\beta_4)\Big)\Big).$$

*For simplicity we can assume $l_1 = l_2 = 1$, then we have*

$$\frac{1}{|\det(\boldsymbol{B})|^2 \exp(\langle \boldsymbol{B}^{-1}\boldsymbol{y},\boldsymbol{l}\rangle)} \int_{t_1}^{t_2} \int_{s_1}^{s_2} \exp(\langle \boldsymbol{B}^{-1}\boldsymbol{x},\boldsymbol{l}\rangle)\mathrm{d}x_1\mathrm{d}x_2$$

$$= \frac{1}{|\det(\boldsymbol{B})|^2 \exp(\langle \boldsymbol{B}^{-1}\boldsymbol{y},[1,1]\rangle)(\beta_1 + \beta_3)(\beta_2 + \beta_4)}\Big(\exp\Big(t_2(\beta_1 + \beta_3) - \exp\Big(t_1(\beta_1 + \beta_3)\Big)$$

$$\times \Big(\exp\Big(s_2(\beta_2 + \beta_4)\Big) - \exp\Big(s_1(\beta_2 + \beta_4)\Big)\Big).$$

**Proof of Theorem 2**

*Proof.* We know that $\int_{\mathbb{R}^2} \frac{k(g^{-1}\boldsymbol{x})\Phi(\boldsymbol{x})}{|\det h|}\mathrm{d}\boldsymbol{x} = \int_{\mathbb{R}^2} k(\boldsymbol{x})\Phi(g\boldsymbol{x})\mathrm{d}\boldsymbol{x}$, then we have

$$\sup_{g'}\big|\big((\mathcal{K}\Phi_1) - \rho\left(g^{-1}\right)(\mathcal{K}\Phi_1)\big)(g')\big|$$

$$= \sup_{g'}\Big|\int_{\mathbb{R}^2} k(\boldsymbol{x})\Phi_1(g'\boldsymbol{x})d\boldsymbol{x} - k(\boldsymbol{x})\Phi_1(gg'\boldsymbol{x})d\boldsymbol{x}\Big|$$

$$\leq \sup_{g'}\int_{\mathbb{R}^2} |k(\boldsymbol{x})||\Phi_1(g'\boldsymbol{x}) - \Phi_1(gg'\boldsymbol{x})|\mathrm{d}\boldsymbol{x},$$

by setting $g'\boldsymbol{x} = \boldsymbol{y}$ for the last term in above we have

$$\sup_{g'}\int_{\mathbb{R}^2} |k(\boldsymbol{x})||\Phi_1(g'\boldsymbol{x}) - \Phi_1(gg'\boldsymbol{x})|\mathrm{d}\boldsymbol{x} \leq \epsilon \int_{\mathbb{R}^2} |k(\boldsymbol{x})|\mathrm{d}\boldsymbol{x} = \epsilon\|k\|_1^{\mathbb{R}^2}.$$

$\square$

**Proof of Theorem 3**

*Proof.* We have

$$\|(\mathcal{K}\Phi_1) * k - \rho(\tilde{h})(\mathcal{K}\Phi_2) * k\|_{\sup}^{G_2} = \sup\Big|\int_{G_2}(\mathcal{K}\Phi_1)(g)k(h^{-1}(g)) - \rho(\tilde{h})(\mathcal{K}\Phi_2)(g)k(h^{-1}g)\mathrm{d}\mu_{G_2}(g)\Big|$$

$$\leq \sup\int_{G_2}\Big|(\mathcal{K}\Phi_1)(g)k(h^{-1}(g)) - \rho(\tilde{h})(\mathcal{K}\Phi_2)(g)k(h^{-1}g)\Big|\mathrm{d}\mu_{G_2}(g)$$

$$\leq \sup\int_{G_2}\Big|(\mathcal{K}\Phi_1)(g) - \rho(\tilde{h})(\mathcal{K}\Phi_2)(g)\Big|\Big|k(h^{-1}(g))\Big|\mathrm{d}\mu_{G_2}(g)$$

$$\leq \epsilon\|k\|_1^{G_2}.$$

The second part of the theorem results by selecting $k = \delta(g - h')$. $\square$

**Proof of Theorem 4**

*Proof.* We know that

$$\left| \int_{G_2} \big( (\mathcal{K}\Phi_1) * k - (\mathcal{K}\Phi_2) * k \big)(h) \mathrm{d}\mu_{G_2}(h) \right| =$$
$$\left| \int_{G_2} \int_{G_2} \big( (\mathcal{K}\Phi_1)(g) k(h^{-1}g) \big) \mathrm{d}\mu_{G_2}(g) \mathrm{d}\mu_{G_2}(h) - \int_{G_2} \int_{G_2} \big( (\mathcal{K}\Phi_2)(g) k(h^{-1}g) \big) \mathrm{d}\mu_{G_2}(g) \mathrm{d}\mu_{G_2}(h) \right|.$$

Then for the second term in the above equation we have and replacing $g$ with $\tilde{h}^{-1}g$ we have

$$\int_{G_2} \int_{G_2} \big( (\mathcal{K}\Phi_2)(g) k(h^{-1}g) \big) d\mu_{G_2}(g) d\mu_{G_2}(h)$$
$$= \int_{G_2} \int_{G_2} \big( (\mathcal{K}\Phi_2)(\tilde{h}^{-1}g) k(h^{-1}\tilde{h}^{-1}g) \big) \mathrm{d}\mu_{G_2}(g) \mathrm{d}\mu_{G_2}(h)$$
$$= \int_{G_2} \int_{G_2} \big( (\mathcal{K}\Phi_2)(\tilde{h}^{-1}g) k((\tilde{h}h)^{-1}g) \big) \mathrm{d}\mu_{G_2}(g) \mathrm{d}\mu_{G_2}(h),$$

if we set

$$\Phi(h) = \int_{G_2} \big( (\mathcal{K}\Phi_2)(\tilde{h}^{-1}g) k((\tilde{h}h)^{-1}g) \big) \mathrm{d}\mu_{G_2}(g),$$

then

$$\int_{G_2} \int_{G_2} \big( (\mathcal{K}\Phi_2)(\tilde{h}^{-1}g) k((\tilde{h}h)^{-1}g) \big) \mathrm{d}\mu_{G_2}(g) \mathrm{d}\mu_{G_2}(h)$$
$$= \int_{G_2} \Phi(h) \mathrm{d}\mu_{G_2}(h) = \int_{G_2} \Phi(\tilde{h}h) \mathrm{d}\mu_{G_2}(h)$$
$$= \int_{G_2} \int_{G_2} \big( (\mathcal{K}\Phi_2)(\tilde{h}^{-1}g) k(h^{-1}g) \big) \mathrm{d}\mu_{G_2}(g) \mathrm{d}\mu_{G_2}(h),$$

therefore,

$$\left| \int_{G_2} \int_{G_2} \big( (\mathcal{K}\Phi_1)(g) k(h^{-1}g) \big) \mathrm{d}\mu_{G_2}(g) \mathrm{d}\mu_{G_2}(h) - \int_{G_2} \int_{G_2} \big( (\mathcal{K}\Phi_2)(g) k(h^{-1}g) \big) \mathrm{d}\mu_{G_2}(g) \mathrm{d}\mu_{G_2}(h) \right|$$
$$= \left| \int_{G_2} \int_{G_2} \big( (\mathcal{K}\Phi_1)(g) k(h^{-1}g) \big) \mathrm{d}\mu_{G_2}(g) \mathrm{d}\mu_{G_2}(h) - \int_{G_2} \int_{G_2} \big( (\mathcal{K}\Phi_2)(\tilde{h}^{-1}g) k(h^{-1}g) \big) \mathrm{d}\mu_{G_2}(g) \mathrm{d}\mu_{G_2}(h) \right|$$
$$= \left| \int_{G_2} \int_{G_2} \big( (\mathcal{K}\Phi_1)(g) - (\mathcal{K}\Phi_2)(\tilde{h}^{-1}g) \big) k(h^{-1}g) \mathrm{d}\mu_{G_2}(g) \mathrm{d}\mu_{G_2}(h) \right|$$
$$= \left| \int_{G_2} \big( (\mathcal{K}\Phi_1) - (\mathcal{K}\Phi_2) \circ \tilde{h}^{-1} \big) * k \, \mathrm{d}\mu_{G_2}(h) \right|$$
$$\leq \int_{G_2} \left| \big( (\mathcal{K}\Phi_1) - (\mathcal{K}\Phi_2) \circ \tilde{h}^{-1} \big) * k \right| \mathrm{d}\mu_{G_2}(h)$$
$$= \left\| \big( (\mathcal{K}\Phi_1) - (\mathcal{K}\Phi_2) \circ \tilde{h}^{-1} \big) * k \right\|_1^{G_2} \leq \epsilon \|k\|_1^{G_2}.$$

$\square$

**Proof of Theorem 5**

*Proof.* We know that

$$\int_{G_2} \Phi([\boldsymbol{x}, \boldsymbol{A}]) k([\boldsymbol{y}, \boldsymbol{B}]^{-1}[\boldsymbol{x}, \boldsymbol{A}]) \mathrm{d}\mu_{G_2}$$
$$= \int_{G_2} \Phi([\boldsymbol{x}, \boldsymbol{A}]) k\big(\boldsymbol{B}^{-1}\boldsymbol{x} - \boldsymbol{B}^{-1}\boldsymbol{y}, \boldsymbol{A}\boldsymbol{B}^{-1}\big) \mathrm{d}\mu_{G_2}.$$

Employing (6) we have

$$\int_{G_2} \Phi([\boldsymbol{x}, \boldsymbol{A}]) k(\boldsymbol{B}^{-1}\boldsymbol{x} - \boldsymbol{B}^{-1}\boldsymbol{y}, \boldsymbol{A}\boldsymbol{B}^{-1}) \mathrm{d}\mu_{G_2}$$
$$= \int_{\mathrm{GL}_2(\mathbb{R})} \int_{\mathbb{R}^2} \Phi[\boldsymbol{x}, \boldsymbol{A}] k(\boldsymbol{B}^{-1}\boldsymbol{x} - \boldsymbol{B}^{-1}\boldsymbol{y}, \boldsymbol{A}\boldsymbol{B}^{-1}) \frac{\mathrm{d}x_1 \mathrm{d}x_2}{|\det(\boldsymbol{A})|} \mathrm{d}\mu_{\mathrm{GL}_2},$$

we also set

$$H_{\Phi,k}(\boldsymbol{A}, \boldsymbol{B}, \boldsymbol{y}) := \int_{\mathbb{R}^2} \Phi[\boldsymbol{x}, \boldsymbol{A}] k(\boldsymbol{B}^{-1}\boldsymbol{x} - \boldsymbol{B}^{-1}\boldsymbol{y}, \boldsymbol{A}\boldsymbol{B}^{-1}) \frac{\mathrm{d}x_1 \mathrm{d}x_2}{|\det(\boldsymbol{A})|}. \tag{9}$$

From separability property of kernel we have $k(\boldsymbol{x}, \boldsymbol{A}) = k_1(\boldsymbol{x}) k_2(\boldsymbol{A})$. As a result

$$\begin{aligned} H_{\Phi,k}(\boldsymbol{A}, \boldsymbol{B}, \boldsymbol{y}) &= \int_{\mathbb{R}^2} \Phi[\boldsymbol{x}, \boldsymbol{A}] k_1(\boldsymbol{B}^{-1}\boldsymbol{x} - \boldsymbol{B}^{-1}\boldsymbol{y}) k_2(\boldsymbol{A}\boldsymbol{B}^{-1}) \frac{\mathrm{d}x_1 \mathrm{d}x_2}{|\det(\boldsymbol{A})|} \\ &= \frac{k_2(\boldsymbol{A}\boldsymbol{B}^{-1})}{|\det(\boldsymbol{A})|} \int_{\mathbb{R}^2} \Phi[\boldsymbol{x}, A] k_1(\boldsymbol{B}^{-1}\boldsymbol{x} - \boldsymbol{B}^{-1}\boldsymbol{y}) \mathrm{d}x_1 \mathrm{d}x_2 \\ &= \frac{k_2(\boldsymbol{A}\boldsymbol{B}^{-1})}{|\det(\boldsymbol{A})|} \Big(\Phi * (k_1 \circ \boldsymbol{B}^{-1})\Big) \\ &= \frac{k_2(\boldsymbol{A}\boldsymbol{B}^{-1})}{|\det(\boldsymbol{A})|} \mathcal{F}^{-1}\Big(\mathcal{F}(\Phi) \mathcal{F}(k_1 \circ \boldsymbol{B}^{-1})\Big), \end{aligned} \tag{10}$$

where $\mathcal{F}(\cdot)$ denotes the Fourier transform. The next step is to find an explicit form for the Fourier transform. We can apply the result from (Bracewell et al., 1993). Assume that $\mathcal{F}(K_1) = \widehat{K}_1(\boldsymbol{u})$ and $\mathcal{F}(\Phi) = \widehat{\Phi}(\boldsymbol{u})$ then we have

$$H_{\Phi,K}(a, b, c, d, \boldsymbol{B}, \boldsymbol{y}) = \frac{K_2(\boldsymbol{A}\boldsymbol{B}^{-1})}{|\det(\boldsymbol{A})||\det(\boldsymbol{B}^{-1})|} \mathcal{F}^{-1}\Big(\widehat{\Phi}(\boldsymbol{u}) \widehat{K}_1(\boldsymbol{B}^{\top}\boldsymbol{u})\Big).$$

Now we use decomposition of $\mathrm{GL}_2(\mathbb{R})$ as $K_0 \ltimes H(1,0)$ in (Milad & Taylor, 2023; Schindler, 1993).

**Proposition 1** (Proposition 5.1 of (Milad & Taylor, 2023)). *If $\boldsymbol{A} = \begin{pmatrix} a & b \\ c & d \end{pmatrix} \in \mathrm{GL}_2(\mathbb{R})$, then $\boldsymbol{A}$ can be uniquely decomposed as the product $\boldsymbol{A} = \boldsymbol{M_A}\boldsymbol{C_A}$ with $\boldsymbol{M_A} \in K_0$ and $\boldsymbol{C_A} \in H_{(1,0)}$. In fact*

$$\boldsymbol{M_A} = \begin{pmatrix} s & -t \\ t & s \end{pmatrix}, \quad \text{with} \quad s = \frac{d(ad - bc)}{b^2 + d^2}, t = \frac{-b(ad - bc)}{b^2 + d^2},$$

*and*

$$\boldsymbol{C_A} = \begin{pmatrix} 1 & 0 \\ u & v \end{pmatrix}, \quad \text{with} \quad u = \frac{cd + ab}{(ad - bc)}, v = \frac{b^2 + d^2}{(ad - bc)}.$$

*This factorization leads to a parallel factorization of $G_2$.*

Consider the one to one transform between $H$ and $H^*$ so that $H^*(s, t, u, v, \boldsymbol{B}, \boldsymbol{y}) := H_{f,k}(a, b, c, d, \boldsymbol{B}, \boldsymbol{y})$, where $a = s - ut$, $c = t + us$, $b = -t/v$, and $d = s/v$. Employing the above proposition and Equation (7) we can write

$$H'(\boldsymbol{B}, \boldsymbol{y}) = \int_{\mathrm{GL}_2(\mathbb{R})} H_{\Phi,k}(\boldsymbol{A}, \boldsymbol{B}, \boldsymbol{y}) \mathrm{d}\mu_{\mathrm{GL}_2}$$

$$= \int_{\mathrm{GL}_2(\mathbb{R})} H^*(s(a, b, c, d), t(a, b, c, d), u(a, b, c, d), v(a, b, c, d), \boldsymbol{B}, \boldsymbol{y}) \mathrm{d}\mu_{\mathrm{GL}_2}.$$

Therefore, we obtain

$$\int_{\mathrm{GL}_2(\mathbb{R})} H_{\Phi,k}(\boldsymbol{A}, \boldsymbol{B}, \boldsymbol{y}) \mathrm{d}\mu_{\mathrm{GL}_2} = \int_{K_0} \int_{H_{(1,0)}} H^*(s, t, u, v, \boldsymbol{B}, \boldsymbol{y}) |v| \mathrm{d}\mu_{H_{(1,0)}} \mathrm{d}\mu_{K_0},$$

as $\det(\boldsymbol{C_A}) = |v|$. Then we define

$$H^*(s, t, \boldsymbol{B}, \boldsymbol{y}) = \int_{H_{(1,0)}} H^*(s, t, u, v, \boldsymbol{B}, \boldsymbol{y}) \det(\boldsymbol{C_A}) \mathrm{d}\mu_{H_{(1,0)}}(u, v)$$

$$= \int_{G_1} H^*(s, t, u, v, \boldsymbol{B}, \boldsymbol{y}) \det(\boldsymbol{C_A}) \mathrm{d}\mu_{G_1}(u, v)$$

$$= \int_{\mathbb{R}} \int_{\mathbb{R}} H^*(s, t, u, v, \boldsymbol{B}, \boldsymbol{y}) \frac{\mathrm{d}u \mathrm{d}v}{|v|}.$$

The next step is to compute integration of $H^*(s, t, \boldsymbol{B}, \boldsymbol{y})$ over $K_0$, which is equal to

$$\int_{\mathbb{R}} \int_{\mathbb{R}} H^*(s, t, \boldsymbol{B}, \boldsymbol{y}) \frac{\mathrm{d}s \mathrm{d}t}{s^2 + t^2}. \tag{11}$$

$\square$

