# OpenReview forum: "Affine Invariance in Continuous-Domain Convolutional Neural Networks"
_TMLR — Rejected by TMLR_

### Review · Reviewer_G19Z · 2026-03-17

**Summary Of Contributions:**

This paper proposes a neural network architecture that is equivariant to 2d affine group. The proposed model consists of lifting layer and group convolution layer that requires integral over group. The author proposes to calculate this integral by exploiting specially designed convolutional kernel and fourier transform. The author also theoretically analyzed the robustness of the group equivariant neural networks. Experimentally, the proposed model demonstrates better robustness to input transformation than non-equivariant model.

**Audience:**

Yes

**Audience Explanation:**

The author proposes a novel affine equivariant neural network structure. It would be better to compare the proposed model with existing work as written in requested changes section.

**Broader Impact Concerns:**

Since this is algorithmic paper, I think possibility of ethical problem is small.

**Claims And Evidence:**

No

**Claims Explanation:**

As for the theory part, given the equivariance of the group convolution and lifting layer, I think the proposed results are the direct consequence of existing mathematical result about the operator norm of convolutional operator. Also, the relationship between the theoretical result and the proposed layer is unclear.

As written in request change section, it is difficult to understand the explicit form of the proposed layer calculation and model structure.

The discussion and experimental comparison to existing work using existing benchmark is insufficient.

**Requested Changes:**

Major

There exist work about equivariant neural networks e.g. “Enabling Equivariance for Arbitrary Lie Groups” [MacDonald et al., CVPR 2022], "Affine Equivariant Networks Based on Differential Invariants" [Li et al., CVPR 2024]. The former constructs similar layer formulation where monte carlo integration over group is used instead of fourier transformation in the proposed method. It would be better to cite these papers, discuss their relationship and experimentally compares to the work to make the contribution of the paper clearer.

Also, it would be better to use the same experimental benchmark as these existing works e.g. affNIST to farily compare the results.

Though the author provides 1d example in Example 5, it is unclear the final form of the proposed affine equivariant layer. It would be better to give the clear formulation of the proposed convolutional layer: shape of the kernel function and it’s learnable parameters, the output of integral calculation and training method.

It would be better to add discussion why existing CNN works better when the sample size is large in Figure 4.

Minor

As for Def 13 and following definitions, I think $\|\|\cdot\|\|_\infty$ would be better than $\|\|\cdot\|\|_1$ to denote uniform norm.

I think the formulation in Theorem 2 is incorrect. I think it’s correct to use different variable for $g$ in $\rho$ and $g$ for input of the function $\Phi$.

It would be better to add equation number to all the equation.

---

> ### Author Response · Authors · 2026-04-12
>
> Dear Reviewer,
>
> Thank you for your thoughtful and constructive feedback. We have carefully addressed all of your comments in the revised manuscript. Below we provide point-by-point responses to your major concerns.
>
> -We thank the reviewer for this suggestion. In the revised manuscript, we added and discussed both MacDonald et al. (2022) and Li et al. (2024) in the related work section. We clarify that our approach differs in that it avoids Monte Carlo integration over the group and instead reduces computation to Euclidean-space operations.
>
> -We thank the reviewer for this suggestion. In the revised manuscript, we have included experiments on the affNIST benchmark in addition to the synthetic and MNIST datasets. This allows for a more direct and fair comparison with existing affine-equivariant methods.
>
> -We thank the reviewer for this comment. In the revised manuscript, we added a clear description of the practical form of the affine-equivariant layer in the implementation section. In particular, we now explicitly describe the kernel shape (a fixed bank of $3\times3$ filters), the discrete approximation of the group convolution via a finite set of affine transformations, the resulting feature maps after projection, and the training procedure. We also clarify that, in the current implementation, only the final classifier is trainable.
>
> -We thank the reviewer for this suggestion. We have added a brief explanation discussing the effect of sample size and the trade-off between built-in invariance and data-driven learning.
> Minor:
>
> -We thank the reviewer for the suggestion. We use the $\ell_1$ norm intentionally, as it aligns with the integral formulation of convolution and supports the stability analysis. While the sup-norm is a valid alternative, the $\ell_1$ norm is more natural for our framework.
>
> -We thank the reviewer for the statement. We fixed Theorem 2 formulation.
>
> We hope these revisions have significantly improved the clarity, readability, and reproducibility of the manuscript. We are grateful for the reviewer’s insightful comments, which have helped us strengthen the paper.

---

### Review · Reviewer_b4LK · 2026-03-28

**Summary Of Contributions:**

This work proposes a group convolutional network to achieve affine invariance for image signals. It introduces a lifting layer that maps 2D images into the Lie group space, followed by group convolutions within this space, and a projection layer that produces outputs invariant to affine transformations. Compared to standard CNNs, the proposed model is inherently robust to arbitrary affine transformations without relying on data augmentation. The work also provides theoretical guarantees for the invariance and stability of the proposed layers. The simulated experiment demonstrates improved accuracy in low-data regimes.

**Audience:**

Yes

**Audience Explanation:**

The work provides a theoretical analysis for an important problem in current neural networks to achieve affine invariance.

**Claims And Evidence:**

No

**Claims Explanation:**

The work theoretically proof the affine invariance and the stability in proposed GCNN for a class of invertible 2x2 transformations. However, the experimental section does not sufficiently support or validate these theoretical findings. Moreover, the experiments are limited in scope of binary images with a synthetic affine transformation, and do not demonstrate the method’s effectiveness in practical settings.

My concerns comes from the following aspects:
• MNIST is a binary image dataset, where the authors can apply the defined the lifting function for Kronecker delta kernel or the step function in Example 4 and 5. For other data types (natural image, or beyond images), how to define the kernel function?
• Are pixels that fall outside the image boundaries after transformation zero-padded? If so, the operation is not necessarily invertible. In MNIST case, the transformed image still appears naturally but for natural images, this assumption might fail.
• No numerical stability analysis is performed to support the major theoretical contribution of the work.
• Any code for the experiment section?
• In Figure 4, the GCNN underperforms compared to the CNN for certain affine transformations. Could the authors clarify the reasons for this behavior? As presented, the claim that "our proposed group convolution network outperforms conventional CNNs" does not appear to hold universally. The expressiveness of GCNN is not discussed comparing to CNN.

The paper discusses the application to image in particular, but the affine invariance is of interest of a broader class of problems. How would the theoretical finding change accordingly.

**Requested Changes:**

1. Could the author clarify what parameters are trainable explicitly? The lifting operator and the projection operators are pre-defined and the convolution kernel in the lie group space is trainable?
2. How to carefully define a kernel function for lifting operator for signals beyond step functions?
3. Section 4.1 discusses the motivation of the proposed method. Moving this to the Introduction would help readers better understand the context and significance of the work.
4. Additional experiments would strengthen the practical impact of the paper. I suggest including experiments on non-binary images (e.g., natural images) or on time-sequential data.
5. A more comprehensive literature survey of recent related work would be helpful. In the last paragraph of page 2, the statement “Previous attempts have been made to investigate spaces that maintain affine-equivariance, but they are restricted to strict conditions, such as cases where the determinant equals 1” is made without references.
6. Include numerical stability analysis of the model, e.g., showing how the distance is affected by random perturbations and P.
7. Notations should be clearly defined when first introduced, to ensure readability for a broader machine learning audience. For instance, clarify \rho in definition 15, and the norms applied to K1 and K2 in Theorem 1.
8. Could the authors add explanations why Theorem 1 does not directly depends on detP?
8. Figure 2 needs more detailed explanations for clarity.

---

> ### Author Response · Authors · 2026-04-12
>
> Dear Reviewer,
>
> Thank you for your thoughtful and constructive feedback. We have carefully addressed all of your comments in the revised manuscript. Below we provide point-by-point responses to your major concerns.
>
> 1- In our implementation, the lifting operator (affine transformations) and projection operator (averaging over transformations) are fixed. The convolution kernels are also fixed and shared across transformed inputs. Consequently, the only trainable parameters are those of the final fully connected classifier, optimized via backpropagation.
>
> This design isolates the effect of affine-aware representations. The framework, however, naturally supports learnable group convolution kernels, which we identify as future work.
>
> 2- The lifting operator is defined via affine coordinate transformations rather than an explicit kernel. For general signals, we implement this using interpolation (bilinear interpolation), which provides a well-defined extension beyond step functions.
>
> 3- We added a short paragraph before contributions list.
>
> 4- The experimental section has been fully revised, and we now include evaluations on real datasets (MNIST and affNIST) in addition to the synthetic experiments. These results demonstrate that the proposed method generalizes beyond binary toy data and remains effective under realistic affine distortions.
>
> 5- We thank the reviewer for this observation. In the revised manuscript, we expanded the related work discussion to include more recent literature, in particular recent approaches on arbitrary Lie-group equivariance and affine-equivariant constructions. We also removed the unsupported statement and replaced it with a more precise discussion supported by appropriate references.
>
> 6- We thank the reviewer for this suggestion. We agree that a dedicated numerical stability analysis under random perturbations would be valuable. In the revised manuscript, we strengthened the empirical evaluation on affNIST and MNIST and clarified the connection between the theoretical stability result and the observed robustness of the model. A more detailed perturbation-based numerical stability study is an important direction for future work.
>
> 7- We thank the reviewer for this helpful suggestion. In the revised manuscript, we have clarified the notation throughout the paper. In particular, we now explicitly define $\rho$ at its first occurrence and specify the norms used in Theorem 1 for $K_1$ and $K_2$.
>
> 8- We thank the reviewer for this insightful comment. While $\det P$ does not appear explicitly in the statement of Theorem 1, its effect is implicitly accounted for through the normalization introduced in the lifting step. This dependence is further clarified in Remark 1 and illustrated in Example 4, where the role of $\det P$ in scaling the transformed signal is discussed. We have added a brief clarification in the revised manuscript to make this connection more explicit.
>
> 9- we added this paragraph:
>
> Figure~2 indeed illustrates how the proposed framework captures affine invariance using simple functions. The purpose of this example is not to restrict the analysis to simple signals, but rather to demonstrate that the affine invariance of more complex shapes and curves can be understood through their decomposition into simpler components. In this sense, the figure provides a conceptual bridge between the theoretical formulation and practical signals, showing how the invariance properties extend naturally from simple functions to general geometric structures.
>
> We hope these revisions have significantly improved the clarity, readability, and reproducibility of the manuscript. We are grateful for the reviewer’s insightful comments, which have helped us strengthen the paper.

---

### Review · Reviewer_hwcF · 2026-03-29

**Summary Of Contributions:**

**Summary:**
The authors study how group-convolutional neural networks can be built to be equivariant with respect to affine transformations, and introduce a new type of affine-equivariant network. They further theoretically provide bounds on how much group-convolutions expand distances between affine-transformed functions. They test their new model on a small synthetic dataset with a small number of hand-picked affine transformations.

**Strengths:**
- The thourough preliminaries section is appreciated and provides a good review for people new to the field. This is a valuable contribution to the community.
- The authors introduce something similar to a 'steerable' convolution for affine transformations, something which I think is valuable to have written explicitly (although technically it may be already captured by prior work, I am not aware of this specific form taking advantage of sums of separable kernels).

**Weaknesses:**
- The introduction is relatively poorly written, overly verbose, yet technically imprecise. The paper would be better suited if the length of the introduction were shortened.
- The organization of Section 4 is also somewhat confusing, it is not clear why the final theorem is stated first, this makes it a bit harder to read. Figure 2 is never described in the text.
- In the end the solution to the affine-equivariant layer is hard to parse from the text.
- The paper only has a single highly synthetic toy dataset. It is not clear why the method would not at least be tested on MNIST. Without variability of digit shape, the dataset is inherently extremely limited and thus overfitting is a serious concern.
- The experiment details are not sufficiently explained.
	- For example, what exactly is the network architecture? How many layers, how many channels, what activation functions?
	- The construction of the test set is never described. The method by which the translation parameter is selected is never described.
	- The training procedure (learning rates, epochs, etc.) is never described.
	- More importantly, how is the G-CNN constructed exactly? The preceeding theory section is highly abstract and does not specify a concretly implementable model. There are many indefinite integrals which are likely somehow approximated in the experiments.
	- In the current manuscript the experiments are not sufficiently described for readers to have faith in the results.
- The experiment design is impoverished relative to comparable work in the geometric deep learning field. Why do the authors present the performance only for a few hand-selected affine transformations rather than doing a larger search? Or even better, why not construct one large dataset with many affine transformations simultaneously?
- The claim in the conclusion that this work "significantly broaden[s] the class of transformations for which G-CNNs are theoretically justified"  is an overstatement in my opinion. Affine equivariance is already theoretically justified by the original G-CNN work, but this paper provides a concrete instantiation.

**Minor:**
- From the second paragraph of the introduction, where the authors define 'stability under affine transform', the notation $∗$ is not yet defined (presumably meaning convolution).
	- Furthermore, this notion of 'stability' appears to be more accurately described as invariance, why do they authors need to create a new term?
- Typo, page 6: "We also need another layer to again maps to feature maps in ..."
- Typo, theorem 1: "three, lifting, convolutional, and R-projection layer"
- Typo, theorem 2: "let there exists an"

**Questions:**
- Why is Phi suddenly defined over both $x$ and $h$ in equation 3?

**Audience:**

Yes

**Audience Explanation:**

Yes, the theory and method for constructing affine G-CNNs would be of interest to researchers interested in geometric deep learning, and the paper provides a nice preliminary section that would be valuable to the community.

**Claims And Evidence:**

No

**Claims Explanation:**

The conclusion is a bit misleading in terms of the contributions. More importantly however, the experiments are very poorly explained to the point where it is not clear what is being tested. Furthermore, the experiments are severly limited in scope, making them unconvincing and leaving room for many potential confounding factors.

**Requested Changes:**

The authors should significantly improve the description of the experiments and expand their scope. I believe these would be critical for a publication ready paper. For example, the full details of training and architecture, and precisely how the GCNNs are constructed and implemented need to be included. Furthermore, it would be highly beneficial if the scope of the experiments would at least be expanded to MNIST, and the performance of the model averaged of many affine transformations rather than a few hand-selected cases.

---

> ### Author Response · Authors · 2026-04-12
>
> Dear Reviewer,
>
> Thank you for your thoughtful and constructive feedback. We have carefully addressed all of your comments in the revised manuscript. Below we provide point-by-point responses to your major concerns.
>
> Introduction
> We agree with your assessment. The introduction has been entirely rewritten and substantially shortened. It now concisely presents the core problem, our key insight, and the main contributions, while eliminating extraneous discussion. All technical imprecisions have been corrected.
>
> Organization of Section 4
> We thank you for pointing out the confusing structure. To improve readability, we have revised the paragraph preceding Theorem 1 as follows:
>
> - Therefore, our first goal is to establish the stability of the proposed three-layer group-convolutional neural network under affine transformations generated by the general linear group $\mathrm{GL}_2(\mathbb{R})$. For clarity, we first state the main stability result, which summarizes the behavior of the overall construction. The subsequent definitions and examples are then introduced to provide intuition and to make the construction explicit.
>
> We have also added a dedicated clarifying paragraph at the end of Section 4 to make the solution to the affine-equivariant layer easier to parse from the text.
>
> Experimental Details
> We thank the reviewer for this valuable feedback. We agree that the original experimental section lacked sufficient clarity and reproducibility. In the revised manuscript, Section 5 has been substantially expanded and rewritten. It now provides a fully explicit description of the model architecture (including the lifting layer, group convolution, projection, pooling, and classifier), with precise details on the number of kernels, layer dimensions, activation functions, and the structure of the fully connected classifier.
>
> We also clearly describe the training procedure (Adam optimizer, learning rate, batch size, number of epochs, and loss function). To bridge the gap between theory and practice, we explicitly explain that the continuous group convolution over the affine group is implemented via a Monte Carlo approximation of the Haar integral using a finite set of sampled transformations. This approximation is realized through standard convolutional operations on transformed inputs followed by averaging, making the model straightforward to implement.
>
> Furthermore, we have improved the description of the datasets and experimental design. The construction of the synthetic dataset is now specified in detail (including how affine transformations are generated and translation parameters sampled). We have expanded the experimental evaluation to include results on both affNIST and MNIST, reporting quantitative metrics, qualitative results, and a numerical stability analysis. Instead of relying on a few hand-selected transformations, we now use a systematic set of sampled affine transformations.
>
> Finally, we have toned down the claims in the conclusion to more accurately reflect the contribution: our work provides a concrete and practical instantiation of affine-equivariant G-CNNs rather than a fundamentally new theoretical justification of affine equivariance.
>
> Minor Typos
> All typos noted by the reviewer have been corrected.
>
> Equation (3)
> The reviewer correctly identified a notational inconsistency. The symbol $\Phi$ was erroneously reused. We have resolved this by renaming the group feature map to $\Psi$ in Definitions 11 and 12, eliminating any confusion with the input feature map.
>
> We hope these revisions have significantly improved the clarity, readability, and reproducibility of the manuscript. We are grateful for the reviewer’s insightful comments, which have helped us strengthen the paper.

---

### Decision · Action_Editor_3Syd · 2026-05-12

**Recommendation:** Reject

**Additional Comments:**

Although this paper received some recognition for its research objective of designing affine-invariant neural networks and its theoretical rationale, it was ultimately rejected due to a critical lack of experimental evaluation to substantiate the effectiveness of the proposed method. Specifically, the review pointed out that the experiments were limited to simple binary image datasets, failing to demonstrate the claimed broad applicability, and that there was a lack of consistency between the harmonic analysis in the theoretical section and the Monte Carlo integration in the implementation section. Furthermore, the fact that the implementation method is merely a variation of prior research and lacks technical novelty, as well as the fact that the accuracy on the affNIST test is significantly inferior to that of prior research, were considered fatal flaws, leading to the conclusion that the claims cannot be sufficiently substantiated at this stage.

**Audience:**

Yes

**Audience Explanation:**

This is because the paper is highly regarded for providing a useful theoretical analysis of achieving affine invariance, a key challenge in neural networks. Furthermore, the proposal of a novel structure for an affine invariant model and the inclusion of explanations of foundational knowledge that hold educational value for the community are more than enough to capture readers’ interest.

**Claims And Evidence:**

No

**Claims Explanation:**

Multiple reviewers have generally evaluated the theoretical basis and technical accuracy of this paper as sound. However, the experimental evidence supporting the claims is limited to simple binary images, lacking the persuasiveness required to demonstrate broad applicability. Furthermore, reviewers have pointed out a lack of consistency between the theory (harmonic analysis) and the implementation (Monte Carlo integration), as well as a significant degradation in accuracy compared to prior research. Overall, while theoretical accuracy is acknowledged, due to the limited experimental evaluation and inferior performance, it is concluded that, at this stage, there is insufficient clear evidence or persuasiveness to support the claims.